# Modelling the mineralogical composition and solubility of mineral dust in the Mediterranean area with CHIMERE 2017r4

Laurent MENUT[1], Guillaume SIOUR[2], Bertrand BESSAGNET[3], Florian COUVIDAT[3], Emilie JOURNET[2], Yves BALKANSKI[4], and Karine DESBOEUFS[2]

[1]Laboratoire de Météorologie Dynamique, Ecole Polytechnique, IPSL Research University, Ecole Normale Supérieure, Université Paris-Saclay, Sorbonne Universités, UPMC Univ Paris 06, CNRS, Route de Saclay, 91128 Palaiseau, France
[2]Laboratoire Inter-Universitaire des Systèmes Atmosphériques, UMR CNRS 7583, Université Paris Est Créteil et Université de Paris, Institut Pierre Simon Laplace, Créteil, France
[3]Institut National de l'Environnement Industriel et des Risques, Verneuil en Halatte, 60550, Parc Technologique ALATA, France
[4]Laboratoire des Sciences du Climat et de l'Environnement, CNRS, CEA, UVSQ, Gif sur Yvette, France

**Correspondence:** Laurent Menut, menut@lmd.polytechnique.fr

**Abstract.** Modelling of mineral dust is often done using one single mean species. But for biogeochemical studies, it could be useful to access to a more detailed information on differenciated mineral species and the associated chemical composition. The fact to differentiate mineral species would also induce different optical properties and densities, then different radiative impact, transport and deposition. In this study, the mineralogical differenciation is implemented in the CHIMERE regional chemistry-transport model, by using global databases. The results show that this implementation does not change a lot the results in term of Aerosol Optical Depth, surface concentrations and deposition fluxes. But the information on mineralogy, with a high spatial (a few kilometers) and temporal (one hour) resolution, is now available and is ready to be used for future biogeochemical studies.

## 1 Introduction

Mineral dust is a major source of aerosol in the Earth system, (Shao et al., 2011). It is studied for many aspects on how it impacts the Earth radiative system among which its contribution to the budget of surface aerosol concentrations (and then to air quality and health issues, (Morman and Plumlee, 2013)) and to the budget of deposited materials over ocean leading to varying biogeochemistry, (Ravi et al., 2011).

This aerosol may be long-range transported and, many modelling studies are conducted, from the global to the regional scale to better understand its life cycle. Numerous uncertainties remain for all the steps of this life cycle. First, emissions over arid areas depend on soil and surface characteristics as well as near surface wind speed (Alfaro and Gomes, 2001; Menut et al., 2005; Kok et al., 2012). Second, transport and mixing depend on boundary layer turbulence, troposphere properties, density and shape of the aerosol: The way to model fine plumes in transport models remains poorly known. As examples, many studies are dedicated to the transport from Africa to Europe and the composition and pathways of dust plumes are difficult to predict, (Engelstaedter et al., 2006; Bessagnet et al., 2008; Stuut et al., 2009; Menut et al., 2015; Middleton, 2017). Third, mineral

dust will end up being removed from the atmosphere under the effect of dry and wet deposition. If dry deposition is relatively well known, wet deposition remains a very uncertain problem to model, being very sensitive to the precipitation or not, to the altitude of the cloud and to the kind of rain compared to the aerosol size distribution. At present, most of the models can provide deposited maps of mineral dust, but it is generally at low horizontal resolution and for a single species representative of all possible dust composition.

Experimentally, dust deposition over the Mediterranean Sea was studied by Desboeufs et al. (2018) to identify the main sources of measured nutrients and trace metals deposited in Corsica. In several stations in the West Mediterranean Basin, Fu et al. (2017) also studied composition of deposited fluxes using the CARAGA deposition collectors network: the chemical signature of deposited fluxes allowed the authors to distinguish the anthropogenic origin of samples from the Saharan dust sources.

Modelled deposition fluxes are used for biogeochemical studies. Studies are rare and were mainly done at the global scale as in Sokolik and Toon (1999), Balkanski et al. (2007) and Wang et al. (2015). At the regional scale, Richon et al. (2017) used these fluxes over the Mediterranean basin to quantify their impact on plankton productivity. At the global scale, this kind of study was performed, for example, by Mahowald et al. (2005), Landing and Paytan (2010) and Ito and Shi (2016), who used the iron content to obtain more realistic results for their biogeochemical cycles studies. More recently, Hamilton et al. (2019) proposed a specific mechanism to describe iron concentrations, from emissions to deposition. In all these studies, as in the review of Mahowald et al. (2018), it appears that a large part of the biogeochemical results uncertainty is due to the difficulty to predict mineral dust deposition fluxes.

The knowledge of mineral composition may also be a way to improve upon the validation of regional simulations of atmospheric pollutants. In addition to surface mass measurements of non-speciated particulate matter (PM), the co-operative programme for monitoring and evaluation of the long-range transmission of air pollutants in Europe (unofficially 'European Monitoring and Evaluation Programme' as EMEP) proposes surface measurements of calcium aerosol content deposition. Dedicated measurements were also done as presented in Guieu et al. (2010) with the European project ADIOS and in Izana, Tenerife (Spain) Kandler et al. (2007). The comparison between these measurements and model outputs is not straightforward if the aerosol mineralogical composition is not estimated directly at the source. For example, linear relationships were proposed between non sea-salt calcium, $nssCa^{2+}$, surface concentration and the corresponding total mineral dust surface concentration, as $[dust] = \alpha \times [nssCa^{2+}]$ where $\alpha$ is a constant factor. Over western Europe, Putaud et al. (2004) suggest $\alpha$=4.55. Over French forests, Lequy et al. (2013) proposed $\alpha$=33 ($R^2$=0.57) and $\alpha$=5 ($R^2$=0.54) for the sites of Breuil and Hesse, respectively. A large variability is observed between these few estimations highlighting the interest to follow up directly the calcium part of the deposited mineral dust while modelling this aerosol. The mineral composition, with distinct refractive indices for each mineral, is also a way to have more confidence when comparing observed and modelled Aerosol Optical Depth (AOD).

In this study, we present the implementation of the mineralogical composition of mineral dust in the CHIMERE regional chemistry-transport model. In place of a unique dust species (as in all state-of-the-art current models), we calculate the following mineral concentrations: calcite, chlorite, feldspar, goethite, gypsum, hematite, illite, kaolinite, mica, quartz, smectite and vermiculite. Our computation includes the explicit chemical composition and solubility of each mineral. In addition, the

concentrations and deposition fluxes of the following chemical elements: magnesium, iron, phosphorus, aluminium, calcium, silicon, manganese et potassium are modelled. This implementation is done using the existing datasets of Journet et al. (2014), that until now, had only used at the global scale. A simulation is performed for the whole year 2012 over a large domain encompassing Africa and Europe. This geographical domain allows to have the most complete as possible aerosol sources estimation and to reproduce correctly all possible transport pathways from the sources areas to the Mediterranean Sea. Results are presented through a comparison to surface measurements in terms of atmospheric concentrations and deposition fluxes. This mineral dust speciation allows to have more details on our ability to correctly model mineral dust.

The measurements used in this study are described in Section 2. The models used for the mineral dust speciation are described in Section 3. Mineral dust emissions and deposition fluxes calculation are detailed in Section 4. The impact to have mineralogy on the modelled mass in quantified in Section 5, a comparison to available observations is presented in Section 6 and a focus on the modelling of calcium is presented in section 7. The last section presents the conclusions.

## 2 The measurements data

In this study, the model accuracy is quantified using several variables: AOD with the AErosol RObotic NETwork (AERONET) data, particulate matter (PM) surface concentrations and deposition fluxes using the European Monitoring and Evaluation Programme (EMEP) data. Note that a dedicated campaign called ADIOS was performed in 2002 over the Mediterranean Sea, Guieu et al. (2010). In this study, we preferred to model a more recent year, 2012, in order to have more numerous surface stations measurements from the AERONET and EMEP networks.

### 2.1 Aerosol Optical Depth with AERONET data

For the evaluation of the long-range transport of aerosols, including the mineral dust, we use the AERONET photometers measurements to compare the measured and modelled AOD. The aerosol optical properties are compared between observations and model using the AERONET measurements (Holben et al., 2001). The comparison is done using the AOD measured at a wavelength of $\lambda$=550nm and using the level 2 data. The reason for using these data is to quantify whether the model can correctly transport mineral dust from Africa (main emission sources) to the Mediterranean Sea and to Europe, where the surface concentrations are later compared. Note that we use the same stations than those described in Menut et al. (2016). The stations are listed in Table A1 and a map showing their locations in presented Figure A1.

### 2.2 Concentrations and deposition with EMEP data

For the surface aerosol concentrations evaluation, we use the EMEP network providing measurements of $PM_{2.5}$, $PM_{10}$ (aerosol with mean mass median diameter less than 2.5 and $10\mu$m, respectively), gaseous species such as $NO_2$ and $O_3$, and aerosols including nitrates, ammonium and sulphates. The data are stored in the EBAS database and informations about these measurements are available at *https://ebas.nilu.no*. In addition, and to evaluate the realism of the development of the mineralogical

speciation, the non-sea salt calcium concentrations, $nssCa^{2+}$ are compared, as well as its wet deposition. The list of the selected stations is provided in Table A2 as well as a map showing their locations in Figure A2.

## 3 Modelling

Simulations are performed in this study using two regional models: (i) the Weather and Research Forecasting (WRF 3.7.1) model calculates the meteorological variables, (ii) the CHIMERE chemistry-transport model (v2017r4) calculates the fields concentrations of gaseous and aerosols based on the 3D meteorological fields. The simulation domain has a constant horizontal grid size of 60 km × 60 km. The modelled period extends from 1 January to 31 December 2012.

### 3.1 Meteorological modelling

The meteorological variables are modelled with the non-hydrostatic WRF regional model in its version 3.7.1, Skamarock et al. (2007). The global meteorological analyses from the National Centers for Environmental Prediction (NCEP) with the Global Forecast System (GFS) products are used to nudge WRF hourly with pressure, temperature, humidity and wind. In order to preserve both large-scale circulations and small scale gradients and variability, the 'spectral nudging' technique was applied, Von Storch et al. (2000). The model is discretized vertically on 28 levels from the surface to 50 hPa. The Single Moment-5 class microphysics scheme is used, allowing for mixed phase processes and super cooled water Hong et al. (2004). The radiation scheme is RRTMG scheme with the MCICA method of random cloud overlap Mlawer et al. (1997). The surface layer scheme is based on Monin-Obukhov with Carslon-Boland viscous sub-layer. The surface physics is calculated using the Noah Land Surface Model scheme with four soil temperature and moisture layers Chen and Dudhia (2001). The planetary boundary layer physics is processed using the Yonsei University scheme Hong et al. (2006) and the cumulus parameterization uses the ensemble scheme of Grell and Dévényi (2002). The aerosol direct effect is considered using the Tegen et al. (1997) climatology

### 3.2 The chemistry-transport modelling

CHIMERE is a chemistry-transport model allowing the simulation of gaseous and aerosols species concentration fields at a regional scale. It is an off-line model, driven by pre-calculated meteorological fields. In this study, the version 2017r4 described in Mailler et al. (2017) is used. Although the simulation is performed on the same horizontal domain and grid between WRF and CHIMERE, the 28 vertical levels of the WRF simulations are projected onto 20 levels from the surface up to 200 hPa for the CHIMERE model.

A complete chemistry is included in the model, a general description of gaseous and aerosol schemes is provided in Mailler et al. (2017) for this model version, including a detailed description of the aerosol scheme in Couvidat et al. (2018). The chemical evolution of gaseous species is calculated using the MELCHIOR2 scheme. The aerosol size distribution is represented using ten bins, from 40 nm to 40 $\mu$m, in mean mass median diameter as described in Menut et al. (2016) and updated in Mailler et al. (2017).

The photolysis rates are explicitly calculated using the FastJX radiation module (version 7.0b), (Wild et al., 2000; Bian et al., 2002). The modelled AOD is calculated by FastJX for the several wavelengths over the whole atmospheric column. As limit conditions, climatologies from global model simulations are used at the boundaries of the domain. In this study, outputs from LMDz-INCA (Hauglustaine et al., 2014) provided all gaseous and aerosols species, except for mineral dust for which the simulations from the GOCART model are used (Ginoux et al., 2001). Anthropogenic emissions are prescribed from the "Hemispheric Transport of Air Pollution" (HTAP) global database, (Janssens-Maenhout et al., 2015). The vegetation fires emissions are quantified using the APIFLAME model described in Turquety et al. (2014) and used, for example, in Rea et al. (2015). They are calculated based on the MODIS area burned product MCD64 (Giglio et al., 2010).

### 3.3 Calculation of deposition

Aerosols, including mineral dust, may be dry or wet deposited, depending on the meteorology and the surface characteristics. The dry deposition velocity is estimated following Zhang et al. (2001):

$$v_d = v_s + \frac{1}{r_a + r_s} \tag{1}$$

with $v_s$ the settling velocity, $r_a$ the aerodynamical resistance depending on the turbulence close to the surface and $r_s$ the surface resistance, depending on the vegetation type. The aerodynamical resistance $r_a$ depends on several turbulent parameters, such as the Monin-Obukhov length $L$, the friction velocity $u_*$, and the dynamical roughness length $z_{0m}$.

Depending on the atmospheric surface layer stability, $r_a$ is estimated following two ways, depending on the atmospheric stability, as:

$$\begin{cases} r_a(stable) & = \frac{1}{ku_*}\left[\ln\left(\frac{z}{z_{0m}}\right) + 4.7(\zeta_r - \zeta_0)\right] \\ r_a(unstable) & = \frac{1}{ku_*}\left[\ln\left(\frac{z}{z_{0m}}\right) + \ln\left(\frac{(\eta_0^2+1)(\eta_0+1)^2}{(\eta_r^2+1)(\eta_r+1)^2}\right) \\ & \quad + 2\left(\tan^{-1}\eta_r - \tan^{-1}\eta_0\right)\right] \end{cases} \tag{2}$$

where $z_r$ a reference height taken at the middle of the first vertical model layer, $\eta_0 = (1 - 15\zeta_0)^{1/4}$, $\eta_r = (1 - 15\zeta_r)^{1/4}$ and $\zeta_0 = z_0/L$, $\zeta_r = z_r/L$, $k$=0.41 the von Karman constant, $L$ the Monin-Obukhov length. $z_0$, the dynamical roughness length, depends on the fraction of land-use for each category and on the season (see Menut et al. (2013a) for details and values).

The surface resistance $r_s$ for aerosols follows the scheme of Zhang et al. (2001) and is calculated as:

$$r_s = \frac{1}{\epsilon_0 \times u_* \times R_1 \times (E_B + E_{IM} + E_{IN})} \tag{3}$$

with $\epsilon_0$ is a constant set to $\epsilon_0=3$, for all landuse categories, $R_1$ a correction factor describing the relative amount of aerosols sticking at the surface, $E_B$ collection efficiency from Brownian diffusion, $E_{IM}$ collection efficiency from impaction and $E_{IN}$ collection efficiency from interception. The $R_1$ factor is estimated following Slinn (1982):

$$R_1 = \exp(-St^{1/2}), \tag{4}$$

where this factor is applied only for particles with $D_p > 5\mu m$. For Brownian diffusion, the resistance is estimated as:

$$E_B = Sc^{-\gamma} \tag{5}$$

where $\gamma$ is a constant depending on the landuse type. In the model, this constant varies between 0.54 and 0.58. For the impaction, the resistance can have a lot of definitions, depending on the landuse. By default, the resistance value used is:

$$E_{IM} = \left(\frac{St}{\alpha + St}\right)^2 \tag{6}$$

The $\alpha$ values are landuse dependent and are tabulated following Zhang et al. (2001). In this model version, a distinction is made between northern and southern hemisphere, in order to use the correct $\alpha$ for a specific modelled day. For specific vegetation types, the formulation is changed. For high vegetation (such as forests), the resistance proposed by Giorgi (1986) is used:

$$E_{IM} = \left(\frac{St}{0.6 + St}\right)^{3.2} \tag{7}$$

For grassland vegetation, a parameterization is proposed by Davidson et al. (1982):

$$E_{IM} = \frac{St^3}{St^3 + 0.753\ St^2 + 2.796\ St - 0.202} \tag{8}$$

Finally, the collection efficiency from interception is calculated as:

$$E_{IM} = \frac{1}{2}\left(\frac{D_p}{A}\right)^2 \tag{9}$$

with $A$ a characteristic diameter given for landuse and seasonal categories. The settling velocity $v_s$ represents the effect of gravity on particles and is calculated as:

$$v_s = \frac{1}{18}\frac{D_p^2\ \rho_p\ g\ C_c}{\mu} \tag{10}$$

with $\rho_p$, the particle density, $D_p$ the mass median diameter of particles, $C_c$ a slip correction factor accouting for the non-continuum effects when $D_p$ becomes smaller and of the same order of magnitude as the mean free path of air, $\lambda$, Seinfeld and Pandis (1998). $g$ is the gravitational acceleration with $g$=9.81 m s$^{-2}$, $\mu$ the dynamic viscosity (here the air dynamic viscosity is set to $\mu_{air}$=1.8 $\times$ 10$^{-5}$ kg m$^{-1}$ s$^{-1}$). The slip correction factor $C_c$ is estimated as:

$$C_c = 1 + \frac{2\lambda}{D_p}\left[1.257 + 0.4 \, \exp\left(-\frac{1.1D_p}{2\lambda}\right)\right] \tag{11}$$

with $\lambda$ the mean free path of air, in meters, estimated as:

$$\lambda = \frac{2\mu_{\mathrm{air}}}{p\sqrt{\dfrac{8M_{\mathrm{air}}}{\pi RT}}} \tag{12}$$

where $M_{\mathrm{air}}$ is the molecular mass of dry air (here 28.8 g mol$^{-1}$), $T$ the temperature (K), $p$ the pressure (Pa), $\mu$ the air dynamic viscosity and $R$ the universal gas constant.

The aerosols wet deposition calculation is separated between rain and snow. There is also a distinction between the wet deposition in-cloud and below cloud.

For below-cloud scavenging, aerosols are scavenged by raining drops. Following Willis and Tattelman (1989), a polydisperse distribution of raining drops is applied:

$$\begin{aligned} N(R) \quad &= 1.06.10^{14} \times P^{-0.0295}(2R)^{2.16} \\ &\times \exp\left(-5679 \times P^{-0.153}2R\right) \end{aligned} \tag{13}$$

with $P$ the precipitation rate in mm/h and $R$ the radius of the droplet (in m). The flux of deposition is calculated with:

$$F_{bc}^i = c^i \times \sum_R \pi R^2 u_g(R)E(R,r_i)N(R) \tag{14}$$

with $i$ the aerosol species, $r_l$ the radius of the particle (in m), $u_g$ the terminal drop velocity (in m/s), $E(R,r_l)$ the collision efficiency of a particle with a raindrop, $N(R)$ (in m$^{-4}$) the raindrop size distribution.

For below-cloud scavenging of particles by snow, the particles are scavenged by appling the parameterization of Wang et al. (2014). A scavenging coefficient $\lambda_{snow}$ is computed with:

$$log\left(\lambda_{snow}\right) = logA + B \tag{15}$$

with $A$ and $B$ fit function depending on the aerosol mean mass median diameter $D_p$. The flux of deposition is calculated with:

$$F_{in}^i = -\lambda_{snow} \times c^i \tag{16}$$

For in-cloud scavenging is here considered only when a precipitation occurs. The rate of deposition is computed by calculating the rate of impaction between hydrometeors and cloud droplets (assumed to have a diameter of 10 $\mu$m). The rate of scavenging is computed with equations 14 and 16 for $D_p$=10$\mu$m.

## 3.4 The mineral dust flux calculation

In the original version of the model, one unique vertical flux of emissions is estimated for a mean averaged species representing all dust species and elements. Emissions are calculated using the Alfaro and Gomes (2001) scheme, optimized following Menut et al. (2005) and using the soil and surface databases presented in Menut et al. (2013b). Since this latter article was published, several changes were implemented in the emissions scheme.

For the erodibility, the original scheme takes into account "United States Geological Survey" (USGS) land use only. In this model version, we added the database proposed by Beegum et al. (2016) to calculate a new erodibility factor, more related to arid areas. For all model cells considered as 'desert', the MODIS erodibility is used while for all other cells, a constant erodibility factor is applied depending on the USGS land use, as in Menut et al. (2013b). This enables to have a more realistic description of the erodibility in arid areas.

In order to take into account the rain effect on mineral dust emissions limitation, a 'memory' function is added. During a precipitation event, the surface emissions fluxes are set to zero. After the precipitation event, a smooth function is applied to account for a possible crust at the surface (and thus fewer emissions). The complete restart of emissions is obtained 12h after the end of a precipitating event, a timing close to the last results found by Lohou et al. (2014). This function is presented in Mailler et al. (2017) and does not take into account the landuse variability.

## 4 Model changes for mineral dust mineralogy

This section presents all changes made in the CHIMERE v2017r4 model, in order to take into account the calculation of mineral dust mineralogy. We describe methodology which permits to split the vertical flux into contributions of differentiated mineral species. The calculation of the chemical composition of the mineral dust is presented. Boundary conditions and deposition fluxes are also described. The distinction between mineralogical species is estimated after the emissions flux calculations. The mineral properties are considered to have a negligible impact on the emissions flux itself.

### 4.1 Mineralogical species information

In order to split the emission flux into several minerals and chemical elements, additional information is required:

- Soil databases describing the relative part of each mineral and each chemical element in each model grid cell

- The relative part of clay and silt in each grid cell

- For each mineral, its density and refractive index

– The solubility of each chemical elements as a function of each mineralogical species

We describe below how we use information from already known and published databases to gather these data.

### 4.1.1 Silt/clay partition and density

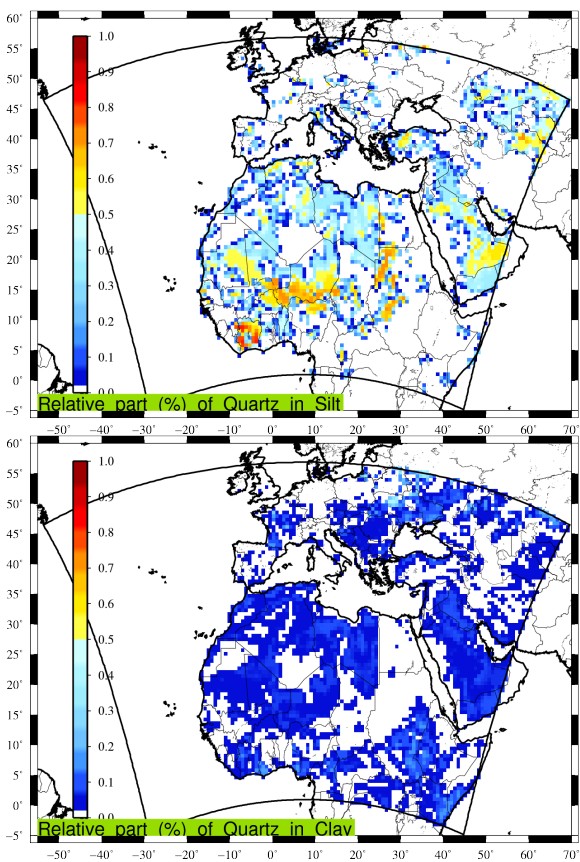

**Figure 1.** *Example of the relative part (0:1) of the quartz mineral in the silt and clay fraction of the soil.*

In order to have information on the dust mineralogical composition, the global databases presented in Journet et al. (2014) are used. These databases are delivered with an horizontal resolution of $0.5^o \times 0.5^o$ and are then interpolated on the model grid used in this study. Data are provided for 12 different species, listed in Table 1. For each mineral, we indicate if it can be found in the clay or silt fraction of the soils. The density of each is also provided: this information comes from several references, including Perlwitz et al. (2015). These values for densities have a non-negligible but not quantified uncertainty. For each mineral, the reference (peer-reviewed publication or internet database) is specified in the Table. Note that for the Mica density, being a group of numerous minerals, the density of muscovite is used.

| Number | Mineral | Silt | Clay | Density |
|--------|---------|------|------|---------|
| 1 | calcite | √ | √ | 2.71 [1] |
| 2 | chlorite | √ | √ | 2.42 [3] |
| 3 | feldspar | √ | √ | 2.68 [1] |
| 4 | goethite | √ | | 4.18 [2] |
| 5 | gypsum | √ | | 2.30 [2] |
| 6 | hematite | | √ | 5.25 [2] |
| 7 | illite | | √ | 2.57 [1] |
| 8 | kaolinite | | √ | 2.63 [1] |
| 9 | mica | √ | | 2.81 [3] |
| 10 | quartz | √ | √ | 2.67 [1] |
| 11 | smectite | | √ | 2.57 [1] |
| 12 | vermiculite | | √ | 2.30 [3] |

**Table 1.** *List of mineral species used in this study (in alphabetical order). Their possible presence in the soil fraction that consists of silt and clay is also indicated. Densities in $g.cm^{-3}$, after (1) Perlwitz et al. (2015), (2) http://www.mindat.org, (3) http://www.engineeringtoolbox.com/mineral-density-d_1555.html. For the Mica density, being a group of numerous minerals, the density of muscovite is used.*

Only five mineralogical species appear in both clay and silt soil fractions: calcite, chlorite, feldspar, goethite and quartz. Two species appear only in the silt fraction: gypsum and mica. Finally, five species are only present in the clay fraction: hematite, illite, kaolonite, smectite and vermiculite. As an example of relative part of a mineral in silt and clay soil, Figure 1 presents the abundance of quartz over the horizontal domain used in this study. It is shown that even if quartz enters in the composition of both silt and clay, it dominates the silt fraction.

### 4.1.2 Refractive indices

In order to be consistent with the approach that includes the mineralogy, we need to know the refractive indices (real and imaginary parts) for each mineral. Values of refractive indices can be gathered from several publications such as Kandler et al. (2007), Utry et al. (2015) and Scanza et al. (2015). As information on the variability of the imaginary part was missing from these references, we use the data of Scanza et al. (2015) in our study and for the following minerals: smectite, illite, hematite, feldspar, kaolinite, calcite, quartz and gypsum. For the goethite, we use the reported in Bedidi and Cervelle (1993). For chlorite and mica, and in absence of accurate information, we use the kaolinite refractive index. For vermiculite, being mainly included in the clay fraction, composed of iron, we use the montmorillonite refractive index (also found in Scanza et al. (2015)).

Values of the real and imaginary parts of the refractive indices are presented in Figure 2 as a function of the solar radiation wavelength ($\mu$m). In addition to the individual minerals, the model species DUST is added. This "mean" species corresponds to what is usually used in models having only one lump species for the mineral dust. All values are reported in Table 2. The

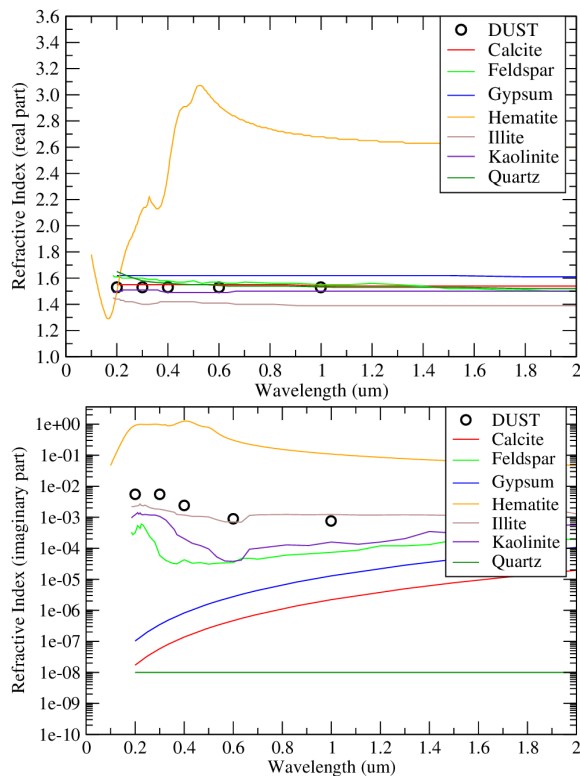

**Figure 2.** *Refractive indices (real and imaginary parts, no unit) for the modelled minerals. "DUST" corresponds to the model species representing a mean mineral dust.*

refractive indices of the mineralogical species modeled here are presented in Table 2. Data are shown for five wavelengths which are the ones chosen to run the FastJX radiative transfer model, implemented online in CHIMERE. For each species and each wavelength, the real and imaginary parts of the refractive index are displayed.

## 4.2 Chemical composition

For each mineral species, we estimate the chemical elements composition for the following 8 elements: magnesium (mg), phosphorus (P), calcium (Ca), manganese (Mn), iron (Fe), aluminium (Al), silicon (Si) and potassium (K). This information was collected from the following previous studies, Kandler et al. (2007), Journet et al. (2014) and Zhang et al. (2015). Values computed in this study are a combination of Journet et al. (2014) and Zhang et al. (2015), and are presented in Table 3. The solubility is also provided as a percentage of each chemical element in each mineral.

## 4.3 Redistribution of emissions and deposition fluxes

The calculation of the mineralogy and chemical elements is divided in two parts:

| λ (nm) | smectite Re | smectite Im(×-1) | illite Re | illite Im(×-1) | hematite Re | hematite Im(×-1) | feldspar Re | feldspar Im(×-1) | kaolinite Re | kaolinite Im(×-1) | calcite Re | calcite Im(×-1) |
|---|---|---|---|---|---|---|---|---|---|---|---|---|
| 200 | 1.57 | 3.98e-03 | 1.44 | 2.29e-03 | 1.49 | 9.13e-01 | 1.61 | 3.16e-04 | 1.50 | 1.20e-03 | 1.55 | 1.72e-08 |
| 300 | 1.56 | 4.37e-03 | 1.40 | 1.82e-03 | 2.12 | 9.73e-01 | 1.60 | 5.89e-05 | 1.51 | 1.07e-03 | 1.55 | 5.81e-08 |
| 400 | 1.54 | 3.16e-03 | 1.42 | 1.18e-03 | 2.37 | 1.25e+00 | 1.58 | 4.27e-05 | 1.49 | 2.04e-04 | 1.55 | 1.38e-07 |
| 600 | 1.52 | 7.59e-04 | 1.41 | 7.08e-04 | 2.93 | 3.12e-01 | 1.57 | 3.47e-05 | 1.49 | 3.80e-05 | 1.55 | 3.60e-07 |
| 999 | 1.51 | 9.33e-04 | 1.39 | 1.20e-03 | 2.68 | 1.11e-01 | 1.56 | 6.61e-05 | 1.50 | 1.23e-04 | 1.55 | 1.40e-06 |

| λ (nm) | quartz Re | quartz Im(×-1) | gypsum Re | gypsum Im(×-1) | vermiculite Re | vermiculite Im(×-1) | chlorite Re | chlorite Im(×-1) | goethite Re | goethite Im(×-1) | mica Re | mica Im(×-1) |
|---|---|---|---|---|---|---|---|---|---|---|---|---|
| 200 | 1.65 | 1.00e-08 | 1.62 | 1.03e-07 | 1.57 | 3.98e-03 | 1.50 | 1.20e-03 | 2.43 | 7.00e-02 | 1.50 | 1.20e-03 |
| 300 | 1.58 | 1.00e-08 | 1.62 | 3.47e-07 | 1.56 | 4.37e-03 | 1.51 | 1.07e-03 | 2.43 | 7.00e-02 | 1.51 | 1.07e-03 |
| 400 | 1.56 | 1.00e-08 | 1.62 | 8.23e-07 | 1.54 | 3.16e-03 | 1.49 | 2.04e-04 | 2.43 | 7.00e-02 | 1.49 | 2.04e-04 |
| 600 | 1.55 | 1.00e-08 | 1.62 | 2.14e-06 | 1.52 | 7.59e-04 | 1.49 | 3.80e-05 | 2.10 | 8.80e-02 | 1.49 | 3.80e-05 |
| 999 | 1.54 | 1.00e-08 | 1.62 | 8.21e-06 | 1.51 | 9.33e-04 | 1.50 | 1.23e-04 | 2.10 | 8.80e-02 | 1.50 | 1.23e-04 |

**Table 2.** *Values of refractive indices (Real and Imaginary parts) selected for the 12 mineralogical species and for the five tabulated wavelengths, λ, required by the FastJX optical properties model.*

1. **The emission fluxes:** As described in section 4, the vertical flux of emitted mineral dust is calculated once, independently of the mineralogy. From this flux, fluxes are calculated for the several mineralogical species. In each grid cell, and for each aerosol bin, instead of having only one "mean" dust species (called DUST), the emission fluxes corresponding to 12 species are computed (all minerals, i.e. DuSmec, DuIlli, and so on) plus the remaning flux that cannot be attributed to a specific mineral (DuOT for "other").

2. **The deposition fluxes:** After emission and transport of the mineralogical species, the calculation is refined for the deposition: fluxes of the chemical elements are estimated for each chemical specie, and their soluble and insoluble parts are computed separately. This distinction is of interest when for example, comparing calcium measurements over land and biogeochemistry over sea. In this case, we calculate the deposition flux for the emitted species (i.e. DuSmec, DuIlli, ..., DuOT) plus the deposition fluxes for their chemical composition (DuFeSo, DuFeIn, etc.).

In addition, one has to note that the boundary conditions for mineral dust are entirely assigned to the "other" species, DuOT. Indeed, having no information from the global model used for the boundary conditions, it was not possible to assign these concentrations to specific minerals. To minimize the impact of this approximation on the boundary conditions, the simulations used in this study are done over a large domain.

### 4.3.1 Emissions fluxes

For the emission fluxes, several specific steps are taken to derive estimates of:

**Chemical composition (%)**

| Mineral | Mg | P | Ca | Mn | Fe | Al | Si | K |
|---|---|---|---|---|---|---|---|---|
| smectite | 1.21 | 0.17 | 0.91 | 0.03 | 2.55 | 8.57 | 27.44 | 0.27 |
| illite | 0.85 | 0.09 | 1.45 | 0.03 | 4.01 | 10.47 | 24.11 | 4.28 |
| hematite | 0.09 | 0.18 | 0.12 | 0.07 | 57.5 | 2.67 | 2.11 | 0.07 |
| feldspar | 0.15 | 0.09 | 3.84 | 0.01 | 0.34 | 10.96 | 25.24 | 5.08 |
| kaolinite | 0.02 | 0.16 | 0.03 | 0.01 | 0.24 | 20.42 | 20.27 | 0.00 |
| calcite | 0.00 | 0.00 | 40.0 | 0.00 | 0.00 | 0.00 | 0.00 | 0.00 |
| quartz | 0.00 | 0.00 | 0.00 | 0.00 | 0.00 | 0.00 | 46.70 | 0.00 |
| gypsum | 0.00 | 0.00 | 23.3 | 0.00 | 0.00 | 0.00 | 0.00 | 0.00 |
| vermiculite | 0.31 | 0.05 | 0.98 | 0.07 | 6.70 | 6.84 | 16.09 | 3.21 |
| chlorite | 9.26 | 0.00 | 0.38 | 0.23 | 12.5 | 6.48 | 15.69 | 0.00 |
| goethite | 0.07 | 0.05 | 0.02 | 0.86 | 62.8 | 0.55 | 0.89 | 0.00 |
| mica | 0.94 | 0.00 | 0.01 | 0.00 | 0.64 | 18.16 | 20.72 | 8.40 |

**Elemental solubility (%)**

| Mineral | Mg | P | Ca | Mn | Fe | Al | Si | K |
|---|---|---|---|---|---|---|---|---|
| smectite | 14.09 | 2.93 | 79.20 | 25.35 | 2.60 | 0.00 | 0.05 | 31.41 |
| illite | 7.80 | 30.58 | 50.96 | 24.93 | 0.17 | 0.15 | 0.05 | 2.87 |
| hematite | 0.00 | 0.00 | 0.00 | 3.39 | 0.01 | 0.00 | 0.00 | 0.00 |
| feldspar | 5.17 | 0.00 | 4.46 | 4.71 | 3.01 | 0.12 | 0.02 | 4.53 |
| kaolinite | 22.32 | 0.00 | 21.97 | 0.00 | 4.26 | 0.38 | 0.37 | 0.00 |
| calcite | 0.00 | 0.00 | 7.00 | 0.00 | 0.00 | 0.00 | 0.00 | 0.00 |
| quartz | 0.00 | 0.00 | 0.00 | 0.00 | 0.00 | 0.00 | 0.0003 | 0.00 |
| gypsum | 0.00 | 0.00 | 0.56 | 0.00 | 0.00 | 0.00 | 0.00 | 0.00 |
| vermiculite | 0.00 | 0.00 | 0.00 | 0.00 | 3.00 | 0.00 | 0.00 | 0.00 |
| chlorite | 0.00 | 0.00 | 0.00 | 0.00 | 2.00 | 0.00 | 0.00 | 0.00 |
| goethite | 0.00 | 0.00 | 0.00 | 0.00 | 0.0006 | 0.00 | 0.00 | 0.00 |
| mica | 0.00 | 0.00 | 0.00 | 0.00 | 0.00 | 0.00 | 0.00 | 0.00 |

**Table 3.** *Generalized chemical compositions and elemental solubility as a percentage of the element contained in the minerals after Paris et al. (2011); Journet et al. (2014); Zhang et al. (2015).*

– the relative part of clay and silt in each model grid cell,

– the relative part of each mineralogical species, as a function of its relative part of clay and silt,

– the rest of the mass, not attributed to clay and silt, thus to a mineralogical species.

The relative part of clay and silt for each mineral depends on the mean mass median diameter of the emitted aerosol. We attempted to follow the formulation proposed in Scanza et al. (2015), with an equation and corresponding results in a Table. Unfortunately, the coding of the proposed formulation provides erroneous values, largely different from the results presented in their Table. Their formulation appears to be numerically not correct, in any case far from the simple goal which is to have a factor giving a smooth transition between 0 and 1. We thus define a new and simplified formulation as:

$$
\begin{cases}
f_{clay}^b & = 1 - \dfrac{c_c}{c_a \times exp(-c_b \times D_p^b)} \\
f_{silt}^b & = 1 - f_{clay}^b
\end{cases}
\tag{17}
$$

with $b$ the aerosol bin number, $D_p$ in $\mu$m, $c_a$=20., $c_b$=1.2 et $c_c$=0.6. These three coefficients were chosen to retrieve results close to the ones presented in the Table of Scanza et al. (2015). The values found with this formulation are displayed in Figure 3. Note that the values correspond to the ten bins defined in this study. The intervals correspond to the values of Scanza et al. (2015).

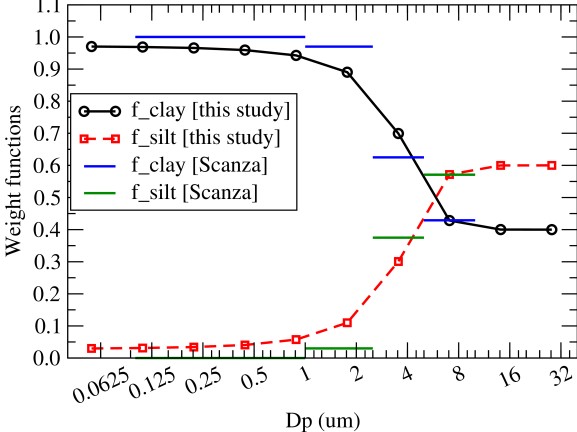

**Figure 3.** *Weight function (0:1) defined to split the relative part of clay and silt as a function of the aerosol mean mass median diameter $D_p$*

Using the relative part of minerals in the clay and silt fraction of the soil, the emissions flux for clay and silt fractions is calculated as:

$$
\begin{cases}
E_{clay}^b & = f_{clay}^b \times E_{tot}^b \\
\\
E_{silt}^b & = f_{silt}^b \times E_{tot}^b
\end{cases}
\tag{18}
$$

with $E_{tot}$ the total vertical emission flux. For each mineral species, and using the values of Table 3, the emission flux is thus estimated as:

$$EF_M^b = E_{clay}^b \times \%_{clayM} + E_{silt}^b \times \%_{siltM} \qquad (19)$$

with $EF_M^b$ the emission flux for the mineral $M$ (i.e Smectite, Illite etc.) and the bin $b$. $\%_{clayM}$ and $\%_{siltM}$ are the percentage of clay and silt, respectively, in the mineral $M$.

Finally, and since the total percentage of all minerals accounted for did not account for 100% of the emitted mass in each model grid cell, the rest of the emitted mass is estimated as:

$$EF_{other}^b = EF_{tot}^b - \sum_{i=1}^{M} \left( EF_i^b \right) \qquad (20)$$

### 4.3.2 Deposition fluxes

The deposition flux of each emitted and transported mineral species is then estimated. In addition, we calculate the flux of the chemical elements pertaining to these mineral species. As described in Table 3, it is possible to assign a relative percentage of each chemical element in each mineral as well as the relative percentage of solubility. The deposition flux for each chemical element is thus calculated as:

$$
\begin{cases}
DF_{Nso}^b & = & \sum_{i=1}^{M} \left( DF_i^b \times \%_{Ni} \times \%_{solubNi} \right) \\
\\
DF_{Nin}^b & = & \sum_{i=1}^{M} \left( DF_i^b \times \%_{Ni} \times (1. - \%_{solubNi}) \right)
\end{cases}
\qquad (21)
$$

with $DF_N^b$ the deposition flux of the chemical element $N$ (i.e. Fe, Ca etc.) for its bin $b$. This flux is splitted between the soluble ($so$) and insoluble ($in$) parts. $\%_{Ni}$ is the percentage of chemical element $N$ in each mineral $M$, $\%_{solubNi}$ the percentage of soluble fraction of chemical element $N$ in each mineral $M$.

### 4.3.3 Boundary conditions

Boundary conditions for mineral dust are calculated using a climatology calculated with the GOCART model, (Ginoux et al., 2001). This climatology was provided by Mian Chin and Paul Ginoux for the CHIMERE validation and distribution to users. The data are freely available on the CHIMERE download web site. The data represent a monthly global climatology simulation of mineral dust with an horizontal resolution of $2.5^o \times 2^o$, Ginoux et al. (2001). The monthly mineral dust concentration fields are an averaged of years 1987, 1988, 1989, 1990 and 1997 and proposed in 7 size bins, later reprojected in the CHIMERE aerosol bins. The use of this climatology in the case of this study has a major weakness, knowing that the data are for the usual mean mineral dust species. The question was thus to choose how to redistribute this mean species into all mineralogical

species. To avoid errors, it was decided to add this contribution into the 'other' species called DuOT. After some test cases, it was shown that the contribution of the boundary conditions was very low compared to the emissions calculated in the modelled domain. The impact of this hypothesis on the results was found to be negligible.

# 5 Impact of the mineralogy on the total modelled mass

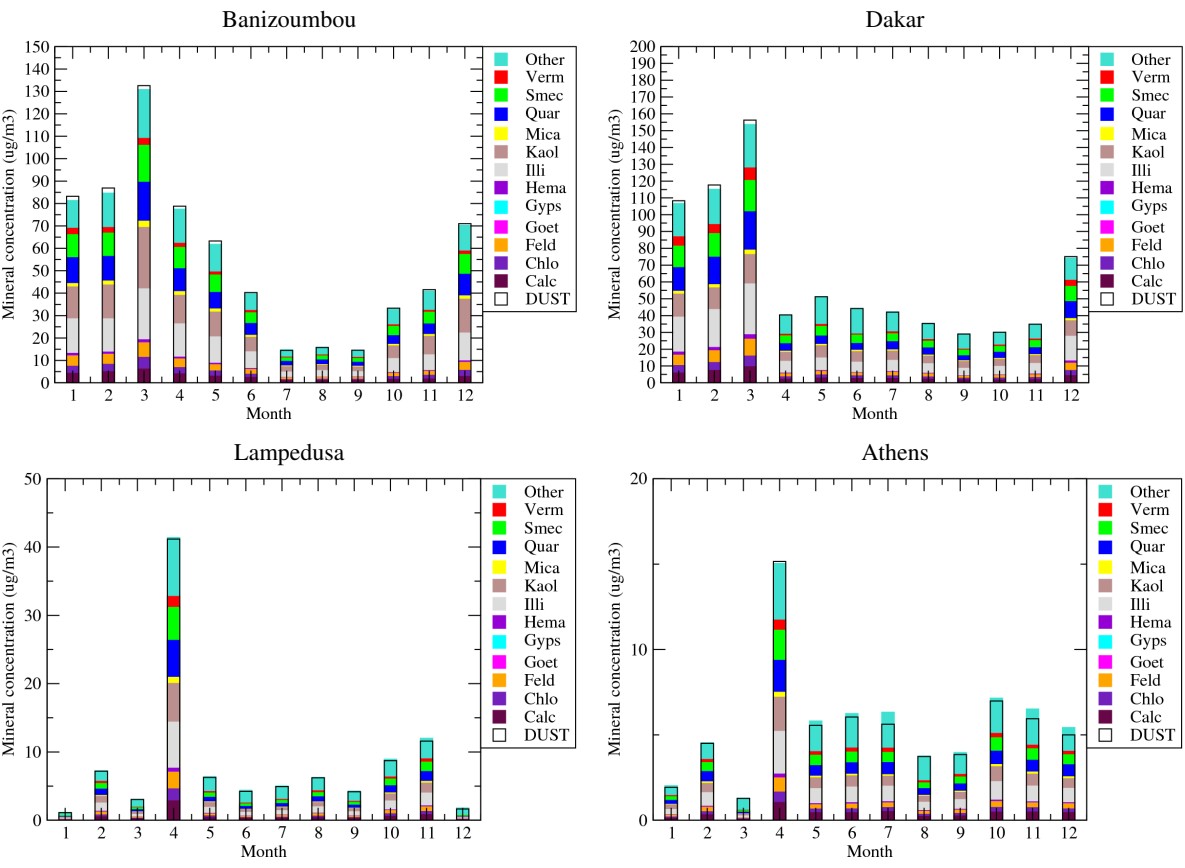

**Figure 4.** *Monthly averaged modelled surface concentrations of the mean DUST species and all mineralogical species ($\mu g\ m^{-3}$).*

This first section of results aims at compare two simulations performed with the WRF-CHIMERE models. The first one, 305 called "DUST", used the mineral dust modelling as presented in Menut et al. (2016), i.e. with only one mean species. The second one, called "MNRLO", used the model developments presented in this study, with different mineralogical species.

The surface concentrations of mineral dust are displayed in Figure 4. Results are presented for the whole year 2012 and for the two simulations (DUST and MNRLO). Having no direct surface measurements of mineral dust, only the model results are presented for intercomparison. Results are shown for four sites, representative of several locations in the model domain:

Banizoumbou and Dakar are close to the mineral dust sources, Lampedusa is an island in the Mediterranean sea and represen-
tative of concentrations after long-range transport, Athens is located in Europe and at the north of the Mediterranean Sea, also
representative of long-range transport of African dust.

   The twelve modelled species of MNRLO are represented as colour bars. The corresponding DUST concentration is presented
as a white box, superimposed to these colours. The time series show simulated concentrations reach a maximum in March and
April over Africa. In Banizoumbou and Dakar, which are close to mineral dust sources, monthly mean surface concentrations
may reach 160 $\mu$g.m$^{-3}$. Over sites more remote from the sources, such as Lampedusa, the mean concentration is around 10
$\mu$g.m$^{-3}$, except in April with the maximum reaches 40 $\mu$g.m$^{-3}$. The same peak is modelled in Athens, but with lower values:
5 $\mu$g.m$^{-3}$ in average during the year and 15 $\mu$g.m$^{-3}$ for the April's peak.

   The relative composition of dust, species per species, in Africa and Europe is close. It means that after emissions (in Africa),
the transport and deposition affect all species in the same manner, even if their densities are different. The main compounds are
Smectite, Quartz, Kaolinite and Illite. These results also show that the decomposition of the mineral dust into several species
has no real impact on the final budget in mass: the sum of all individual mineral species concentrations of MNRLO is close to
the DUST species. This is verified whatever the studied site, near or remote from sources.

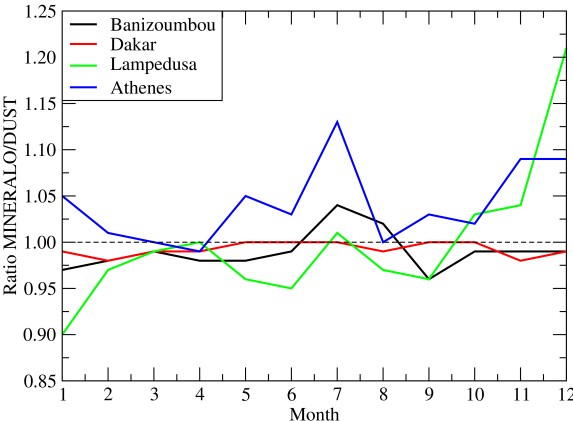

**Figure 5.** *Surface concentrations ratios (ad.) between MNRLO and DUST and for four sites: Banizoumbou, Dakar, Lampedusa and Athens.*

   In order to better quantify the changes in mass between DUST and MNRLO, the ratio of MNRLO/DUST is calculated for
each month. Results are presented in Figure 5 for the four sites presented above. The ratios are varying a lot from month to
month and differently for each site. But, globally, its variations are within an interval from 0.9 to 1.15. It means that whatever
the location and the period of the year, the fact to model explicitly resolves the mineralogical composition induces a maximum
change of $\approx \pm 15\%$. One can also note that for April, when the largest mass are simulated, the ratio is close to 1 at the four
sites: the differences are not linearly dependent on the concentration.

The only parameter affecting the total concentrations is aerosol density. This parameter directly affects dry deposition. The fact that we have a weak difference indicates that the deposition of the aggregated density of the individual mineral species (depending on their relative abundance) is close to the averaged density used for the species DUST alone.

## 6 Model vs observations

In this section, results from the simulations are compared to observations. Mineral dust concentrations being never directly
measured, comparison is achieved on variables linked to it. First, comparison is done with particulate matter surface concentrations (EMEP network). Second, AOD (AERONET photometers) and third, nssCA$^{2+}$ deposition fluxes (EMEP).

### 6.1 Surface concentrations of PM$_{2.5}$

The comparison between surface measurements of PM$_{2.5}$ and the model is presented in Table 4. The stations are located in western Europe and the composition of the particulate matter is a mix between anthropogenic, biogenic, mineral dust and
biomass burning contributions, (Menut et al., 2016). Results are presented as mean values over all stations to have a integrated view of the differences between the two simulations.

| PM$_{2.5}$ | DUST | MNRLO |
|---|---|---|
| R$_s$ | 0.38 | 0.39 |
| $\overline{R_t}$ | 0.37 | 0.37 |
| RMSE | 3.13 | 3.12 |
| bias | 6.23 | 6.19 |

**Table 4.** *Comparison of daily mean PM$_{2.5}$ surface concentrations ($\mu g\ m^{-3}$) between EMEP observations and the CHIMERE model. Results are presented for the spatial correlation $R_s$ between the mean observed and modeled values, and the mean averaged values of temporal correlation, RMSE and bias.*

The error statistics are in-line with the range of what is currently modelled for PM$_{2.5}$ in areas with multiple sources such as the Western Europe. But the most striking point in this table is how similar the two simulations are. The spatial correlation, R$_s$ is 0.38 and 0.39 for DUST and MNRLO, respectively. The score is low and there is no clear improvement using the speciation
of the mineralogy. The same is true for the other statistical values: the averaged temporal correlation, $\overline{R_t}$, is the same. Only the bias is slightly lower for MNRLO with a value of 6.19 (to be compare to 6.23 for DUST), but the difference between the simulations is feable and these differences cannot be considered as significative.

The statistical values are not satisfying: to better understand them, time series are presented, as examples, in Figure 6. Results are presented for Diabla Gora and Harwell. For these two sites, concentrations are maximum in winter. We can see the large
temporal variability of measured and modelled values. Although the model does not always catch the day to day variability, the

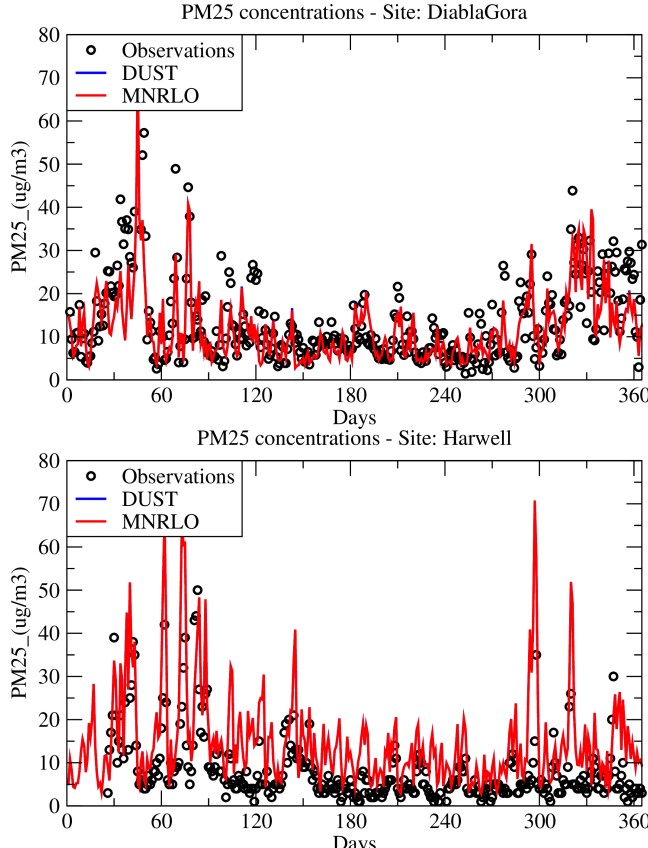

**Figure 6.** *Daily averaged surface concentrations of PM$_{2.5}$ with the EMEP measurements and the two CHIMERE simulations, DUST and MNRLO.*

main tendencies and the background values are correctly captured. In terms of surface concentrations, there is no significant differences between DUST and MNRLO.

## 6.2 Surface concentrations of individual aerosols

More detailed statistical scores are presented in Table 5. The comparison is made between modeled and measured (EMEP network) daily averaged surface concentrations. For each specie, the number of daily data varies from 17 to 83 over Europe. The statistics are expressed with the Root Mean Squared Error (RMSE), the correlation R, the mean fractional bias and error, MFB and MFE, respectively (see, among many others, (Chang and Hanna, 2004; Boylan and Russell, 2006) for the definitions of these metrics).

The analyzed species are calcium (Ca, $\mu$g.m$^{-3}$), sulphate (SO$_4^{2-}$, $\mu$g.S.m$^{-3}$), nitrate (NO$_3^-$, $\mu$g.N.m$^{-3}$), nitrogen oxides (NO$_2$, $\mu$g.m$^{-3}$). For the simulation without mineralogy, the calcium is estimated as $\alpha$DUST with $\alpha$=0.06. For inorganic

| Species | Simulation | $\overline{c_{mod}}$ | $\overline{c_{obs}}$ | RMSE | R | MFB | MFE |
|---|---|---|---|---|---|---|---|
| $\alpha$DUST | DUST | 0.09 | 0.18 | 0.30 | 0.30 | -0.97 | 1.15 |
| $Ca^{2+}$ | MNRLO | 0.06 | 0.18 | 0.30 | 0.26 | -1.29 | 1.39 |
| WCa | DUST | 0.73 | 0.91 | 7.04 | 0.07 | -0.39 | 0.45 |
|  | MNRLO | 0.44 | 0.91 | 6.34 | 0.08 | -0.51 | 0.54 |
| $SO_4^{2-}$ | DUST | 1.52 | 1.89 | 1.62 | 0.40 | -0.14 | 0.53 |
|  | MNRLO | 1.52 | 1.89 | 1.62 | 0.40 | -0.14 | 0.53 |
| $NO_3^-$ | DUST | 3.94 | 2.22 | 4.13 | 0.55 | 0.28 | 0.75 |
|  | MNRLO | 3.95 | 2.22 | 4.13 | 0.55 | 0.28 | 0.75 |
| $NO_2$ | DUST | 7.61 | 6.82 | 7.76 | 0.40 | 0.15 | 0.60 |
|  | MNRLO | 7.60 | 6.82 | 7.76 | 0.40 | 0.15 | 0.60 |
| Mg | MNRLO | 0.07 | 0.12 | 0.15 | 0.50 | -0.68 | 0.90 |
| WMg | MNRLO | 0.15 | 0.59 | 3.76 | 0.02 | -0.49 | 0.53 |

**Table 5.** *Daily mean surface concentrations and wet deposition fluxes of gas and aerosols. Comparisons are presented between EMEP measurements and CHIMERE modelling. Aerosol species are mineral dust, calcium ($Ca^{2+}$, $\mu g.m^{-3}$), sulphate ($SO_4^{2-}$, $\mu g.S.m^{-3}$), nitrate ($NO_3^-$, $\mu g.N.m^{-3}$), nitrogen oxides ($NO_2$, $\mu g.m^{-3}$). In the case of DUST, the calcium concentration is estimated by using the surface concentrations of mineral dust multiplied by a factor $\alpha=0.06$. In case of MNRLO, the calcium and magnesium (Mg, $\mu g.m^{-3}$) is explicitly modeled as well as their respective wet deposition, WCa and WMg ($\mu g.m^{-2}.day^{-1}$).*

species ($SO_4^{2-}$, $NO_3^-$) and $NO_2$, the statistical scores have a satisfying correlation from 0.40 to 0.55 over the period and the domain. The MFB shows an overestimation of $NO_3^-$ and $NO_2$ but an underestimation of $SO_4^{2-}$. For calcium, the use of the mineralogy does not change significantly the results: the correlation is 0.3 without mineralogy and 0.26 with mineralogy, and the bias is increased with mineralogy. The differences between the two simulations are not significant, and the statistical scores are not improved with the explicit calculation of the mineralogy.

### 6.3 Optical depth

We now present the comparison of Aerosol Optical Depth results with AERONET measurements. We want to quantify whether or not the different refractive indices of the individual minerals have an impact on AOD calculation.

First, maps are presented in Figure 7. The top panel displays the mean averaged value of AOD for April 2012 (when the largest surface concentrations were modelled) and for the simulation 'DUST'. Two large areas of high AOD values are modelled in Africa where maximum values reached are larger than 2. The locations correspond to active mineral dust sources. The bottom panel presents the difference AOD(DUST)-AOD(MNRLO). The largest differences appear where there are maximum absolute values of AOD in Sahara and Sahel. Aside from this area, the differences are close to zero and could be considered as non-significant. These results show that the use of speciated dust tends to increase the AOD, but the impact does not affect long-range transported dust.

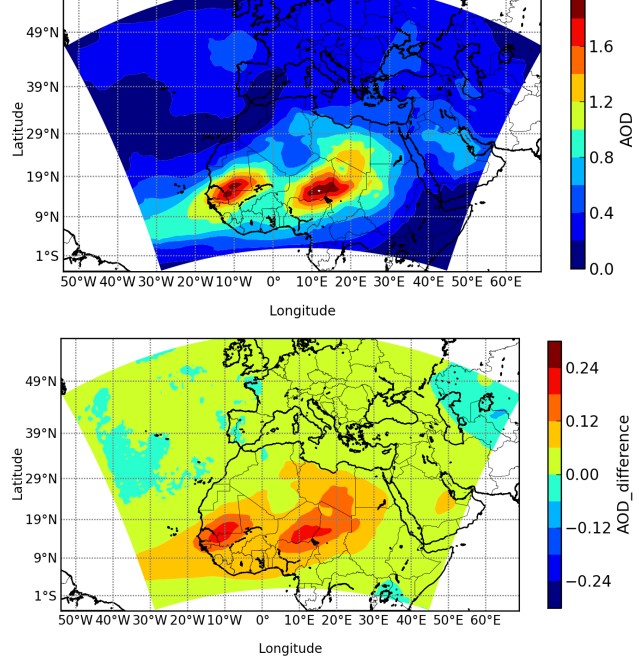

**Figure 7.** *Monthly averaged Aerosol Optical Depth for April 2012 and over the whole modeled domain. (top) AOD absolute values are presented for the simulation DUST. (bottom) The map of difference represents the calculation of AOD(DUST)-AOD(MNRLO).*

Statistical scores are calculated over 32 AERONET stations and are presented in Table 6. As for previous results, differences between the two simulations are small. In terms of yearly mean AOD value, DUST provides higher values than MNRLO, as displayed with Figure 7. There is a positive bias of the simulations compared to the measurements, i.e., the two simulations produce larger AOD than the measurements.

| AOD | DUST | MNRLO |
|---|---|---|
| $R_s$ | 0.95 | 0.94 |
| $\overline{R_t}$ | 0.51 | 0.50 |
| RMSE | 3.58 | 3.49 |
| bias | 0.09 | 0.07 |

**Table 6.** *Comparison between observations (AERONET) and model (CHIMERE) for the daily mean Aerosol Optical Depth (AOD). Results are presented for the spatial correlation $R_s$ between the mean observed and modelled values, and the mean averaged values of temporal correlation, RMSE and bias.*

The bias is lower for MINERAL than for DUST, but the differences are not significative (0.07 versus 0.09). Only the Root Mean Square Error (RMSE) is improved between the two simulations (3.58 for DUST and 3.49 for MNRLO). Finally, the speciation of the dust does not bring a significative improvement on the AOD modelling.

## 7    Modelling of calcium

In this section, modelled calcium contributed to by mineral dust is evaluated through a comparison to measurements. Thus,
only the simulation MNRLO can be compared to measurements.

### 7.1    Deposition fluxes

Simulated monthly mean wet deposition fluxes of $nssCA^{2+}$ and the cumulative precipitations are compared with EMEP measurements in Figure 8. Results are shown for nine European sites located far away from Saharan dust sources. Symbols are used to represent wet deposition, whereas solid lines indicate the values for precipitations.

The precipitations are well captured by the model. The absolute values of precipitations are close between model and measurements. For each site, the seasonal cycle is also well reproduced with a peak in June at Vysokoe and Schmücke, and peaks in July and October at Zingst, amongst other sites.

In contrast, the simulated deposition fluxes underestimate significantly the observed fluxes. While measurements never approach 0 mg.m$-2$, the deposition fluxes simulated by the model are more sporadic and close to 0 for several months.
The measurements exhibit summer maxima, but they are always captured by the model, in some sites, depositions fluxes are simulated in April (Ispra) or in September and October (Brotjacklriegel and Westerland). The capability of a model to simulate the dust cycle contains many processes and, then, many possible errors. Here, the precipitation is correctly represented by the model. It means that the underestimation of modelled wet deposition fluxes compared to the measurements is probably due to other processes than a misrepresentation of the precipitation. This could be the altitude of the precipitating clouds, the
trajectory of dust plumes (missing a station or not), the efficiency of the parameterized scavenging, possible errors on dust size distribution, a too small simulated fraction of $nssCa^{2+}$, among other possibilities.

Statistical results are presented in Table 7 for the 35 EMEP stations available in 2012. They show a large variability between the stations. The modelled values are also clearly underestimated. Independently of this underestimation, the temporal correlation is not good and does not exceed 0.42 (at the DE0044R station).

Since dust plumes are very spatially extensive, there is usually a bias between model and measurements for groups of stations located beneath these plumes. This is not the case here, since there are highly variable biases for nearby stations. The origin of the bias is therefore not due to a 'large scale' error: it is therefore probably not a transport problem. But it may be a precipitation problem, which is often a phenomenon of greater spatial variability on a small scale.

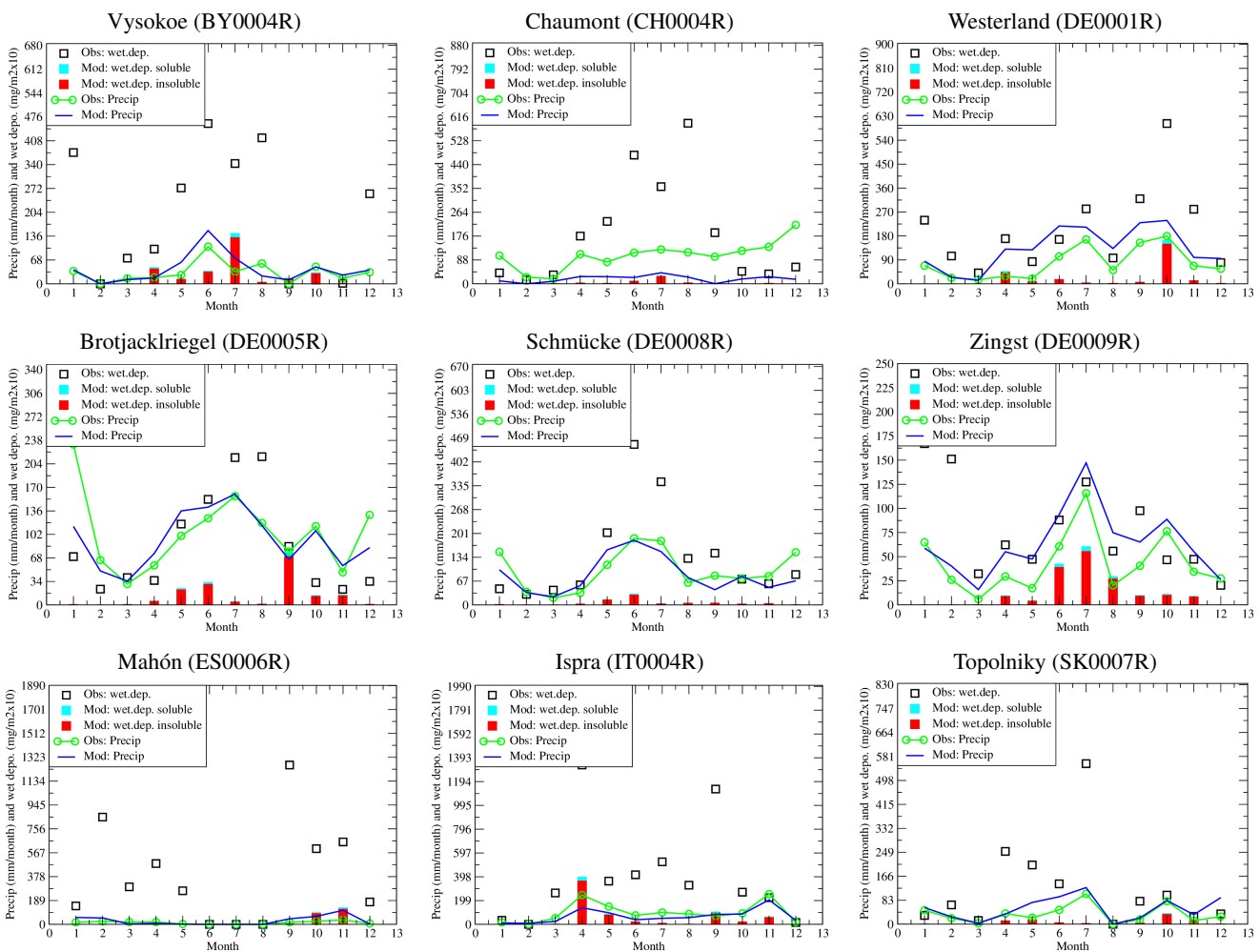

**Figure 8.** *Time series of monthly precipitation rates (mm/month) and nssCa$^{2+}$ wet deposition fluxes (mg/m$^2 \times 10$). The bars represent the monthly deposition fluxes of Ca$^{2+}$, soluble in blue and insoluble in red. The blue line represents the simulated monthly cumulated precipitations. The measurements are represented with symbols. Results are presented for the whole year of 2012 and are observations are accumulated monthly from weekly measurements.*

## 7.2 The nssCA$^{2+}$/dust ratio

The ratio between nssCA$^{2+}$ and total mineral dust concentration is often used to convert measurements of calcium into a total mass of dust. It is used to compare observations to model outputs. Usually, previous authors present ratios of dust/nssCA$^{2+}$. But for low values nssCA$^{2+}$, this ratio may have important values difficult to interpret. Since the goal is to quantify the relative amount of nssCA$^{2+}$ in a total mass concentration of mineral dust, it seems more logical to express the results as a ratio between 0 and 1 with nssCA$^{2+}$/dust.

The explicit modelling of dust mineralogy and chemical composition, allows to plot a map of this ratio, Figure 9. The values represented in the Figure consist of an averaged of all values simulated during the year 2012. Values are in the range between 0 and 0.05 over the whole domain. Maxima are modeled in Africa and the eastern part of the modelled domain. Over western Europe, the values are lower and between 0 and 0.03. Over the Mediterranean Sea, the ratio is relatively homogeneous and with values close to 0.035. Considering this map, it seems clear this is not realistic to use a single and constant value to convert nssCA$^{2+}$ mass measurements into mineral dust.

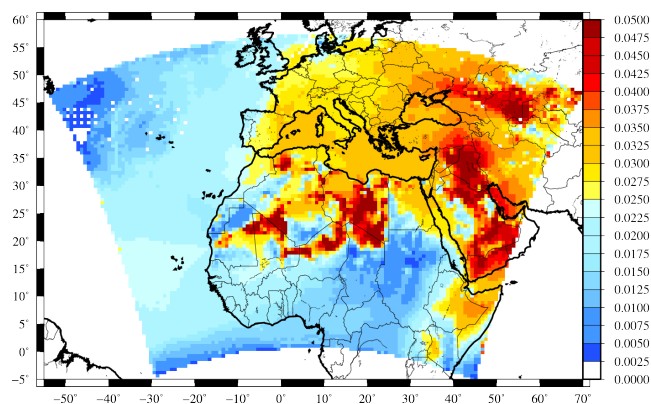

**Figure 9.** *Modeled ratio of nssCA$^{2+}$ on total mineral dust concentrations. The ratio corresponds to the yearly averaged value.*

In order to compare the map of results to previously published values, we report in Table 8 the values found in the literature and the calculation made in this study. The results with the model is close to the values found by Lequy et al. (2013), but only for the Breuil site. But, if we consider there is a strong model bias (as shown in Figure 8) and only for nssCA$^{2+}$ and not the complete mass of mineral dust, then these values should be closer than the ones found by Lequy et al. (2013) (Hesse site) and Putaud et al. (2004).

## 8   Conclusions

The present study consists of the implementation of the mineralogical speciation of dust in the CHIMERE chemistry-transport model. Several databases were implemented and twelve minerals are explicitly treated in ways of emission, transport and deposition in contrast to a single one with a classic approach. A new and simple function is also introduced to correct of the effect of wet sieving and partition accurately the relative part between the silt and the clay fractions. Several motivations justify the need to have dust mineralogical speciation: to better follow the emissions depending on soil type, to better capture the aerosol radiative effect, to better inform biogeochemical models and improve the comparison of deposition fluxes to the available measurements.

We infer that surface concentrations of particulate matter, considered here as PM$_{2.5}$ surface concentrations are close between DUST and MNRLO. On the one hand we would expect this result since the total mass of mineral dust emissions is the same

for DUST and MNRLO. In another hand it can be surprising: since the densities of individual mineralogical species depart from the average density used for the mean DUST species, hence one might have expected to see larger differences during the transport due to differential particle settling between the different minerals. The mean density used for the bulk species DUST is thus well representative of the whole set of mineralogical species. Concerning aerosol optical depth, this study confirmed the

statistical scores when comparing simulated optical depth to retrived ones, but no particular improvements were obtained by using MNRLO in place of the single DUST. Despite the large variability of refractive indices, the calculation based on twelve species did not improved the AOD calculation. Once again, it means that the use of the mean averaged refractive indices seems to be a good proxy of dust aggregates.

Modelled calcium part of the mineral dust was compared to EMEP measurements. Results showed large negative biases.

A major part of mineral dust coming from north Africa, one could have expected the error to increase with the distance from the sources, but it was not the case. For $nssCa^{2+}$ wet deposition fluxes, the modelled values underestimate significantly the measurements. On the other hand, precipitation is reasonably modelled, showing that the problem could come from the representation of the dust plume itself or from additional sources not accounted for but not from the meteorology. Finally, the ratio $nssCa^{2+}$/dust is estimated. Often used as proxy for biogeochemical studies, the implementation of the mineralogy enables

to calculate it explicitly. A yearly averaged map is proposed and for locations where values were proposed in the literature, we showed that our results are fairly close to the observed ones.

These results showed that the implementation of the mineralogical speciation in the model provides additional information for use with biogeochemical modelling and does not change significantly the results in terms of AOD or surface concentrations. A step forward could be to add the Fe anthropogenic emissions in the model and then to have realistic Fe concentrations and

deposited fluxes to make comparisons to measurements as done for example in Ito et al. (2019).

*Acknowledgements.* We acknowledge Jan P.Perlwitz for his very important help to find all informations about the refractive index and densities of the mineralogical species. The EBAS database has largely been funded by the UN-ECE CLRTAP (EMEP), AMAP and through NILU internal resources. Specific developments have been possible due to projects like EUSAAR (EU-FP5) (EBAS web interface), EBAS-Online (Norwegian Research Council INFRA) (upgrading of database platform) and HTAP (European Commission DG-ENV) (import and

export routines to build a secondary repository in support of www.htap.org). Many specific projects have supported development of data and meta data reporting schemes in dialog with data providers (EU) (CREATE, ACTRIS and others).

*Author contributions.* LM and GS developed the code in the CHIMERE model, designed the experiments, performed the simulations, produced Figures and Tables and write the manuscript. BB and FC reviewed and corrected the code and the manuscript, EJ and YB provided the model surface databases, essential for the calculations and primarily developed for the LMDz-INCA model, and KD updated the informations

necessary in the Tables of the mineral composition and elemental solubility. All co-authors contributed to the final version of the manuscript.

*Competing interests.* The authors declare that they have no conflict of interest.

*Data availability.* The model version is available upon request to the first author.

# Appendix A:  Coordinates of the measurements sites

Coordinates of the measurement sites are presented in this Appendix.

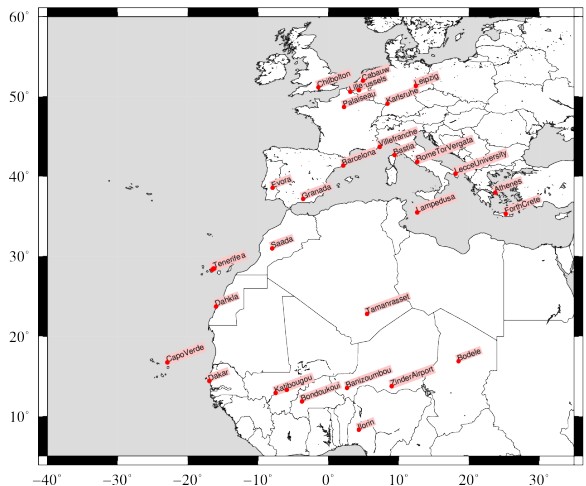

**Figure A1.** *Maps of AERONET sites for the AOD measurements.*

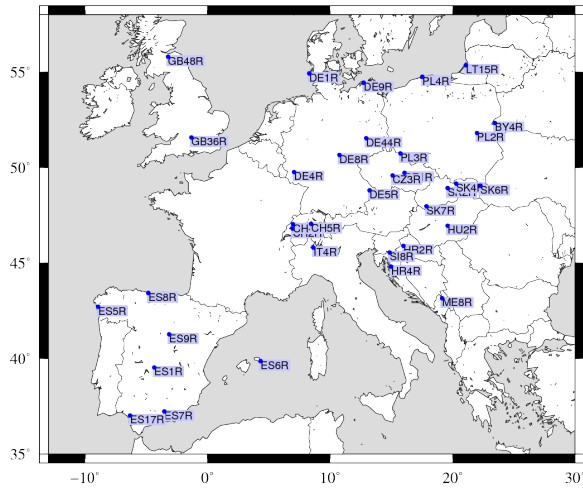

**Figure A2.** *Maps of EMEP sites for the PM$_{2.5}$ and calcium measurements.*

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

| Weekly mean deposition fluxes of nssCA$^{2+}$ (mg/m$^2$/week) | | | | | |
|---|---|---|---|---|---|
| Site | Obs | Mod | $R_t$ | RMSE | bias |
| CH0002R | 5.37 | 0.12 | 0.2 | 8.95 | -5.24 |
| CH0004R | 5.12 | 0.12 | -0.03 | 9.4 | -5 |
| CH0005R | 6.92 | 0.09 | 0.22 | 12.52 | -6.83 |
| CZ0001R | 4.21 | 0.42 | 0.19 | 6.86 | -3.79 |
| DE0001R | 5.61 | 0.6 | 0.29 | 6.7 | -5.01 |
| DE0004R | 3.17 | 0.81 | 0.35 | 5.36 | -2.36 |
| DE0005R | 2.36 | 0.4 | 0.31 | 3.29 | -1.96 |
| DE0008R | 3.63 | 0.16 | 0.38 | 5.63 | -3.47 |
| DE0009R | 2.14 | 0.4 | 0.1 | 2.76 | -1.74 |
| DE0044R | 2.09 | 0.17 | 0.42 | 2.45 | -1.92 |
| SK0007R | 4.86 | 0.3 | 0.27 | 8.01 | -4.56 |
| BY0004R | 3.66 | 0.47 | -0.06 | 6.44 | -3.19 |
| CZ0003R | 3.92 | 0.16 | 0.41 | 5.74 | -3.76 |
| ES0001R | 3.69 | 0.08 | 0.06 | 8.59 | -3.61 |
| ES0005R | 2.32 | 0.02 | 0.08 | 2.98 | -2.3 |
| ES0006R | 9.28 | 0.49 | -0.07 | 12.23 | -8.78 |
| ES0007R | 14.64 | 0.22 | 0.41 | 18.21 | -14.42 |
| ES0008R | 3.63 | 0.07 | 0.29 | 4.72 | -3.57 |
| ES0009R | 9.4 | 0.32 | 0.21 | 19.93 | -9.08 |
| ES0017R | 2.99 | 0.21 | 0.4 | 5.15 | -2.78 |
| GB0036R | 0.85 | 0.07 | 0.21 | 1.56 | -0.78 |
| GB0048R | 0.49 | 0.01 | 0.11 | 0.76 | -0.48 |
| HR0002R | 6.53 | 0.29 | 0.3 | 9.31 | -6.24 |
| HR0004R | 16.43 | 0.6 | 0.14 | 68.01 | -15.84 |
| HU0002R | 4.23 | 0.1 | 0.26 | 6.14 | -4.13 |
| IT0004R | 5.10 | 0.74 | 0.31 | 9.05 | -4.36 |
| LT0015R | 2.62 | 0.02 | 0.23 | 3.84 | -2.6 |
| ME0008R | 22.82 | 1.19 | 0.32 | 32.82 | -21.63 |
| PL0002R | 1.37 | 0.1 | 0.33 | 2.36 | -1.27 |
| PL0003R | 2.01 | 0.03 | 0.05 | 2.73 | -1.98 |
| PL0004R | 0.69 | 0.09 | 0.22 | 0.97 | -0.61 |
| SI0008R | 3.29 | 0.47 | 0.33 | 6.27 | -2.83 |
| SK0002R | 1.28 | 0.12 | 0.31 | 2.25 | -1.16 |
| SK0004R | 1.44 | 0.17 | 0.4 | 2.52 | -1.27 |
| SK0006R | 1.74 | 0.18 | 0.38 | 2.75 | -1.56 |
| Average | $R_s$= 0.34 | | 0.24 | 8.78 | -4.57 |

**Table 7.** *Comparisons between observations (EMEP) and model (CHIMERE) for the weekly mean deposition fluxes of nssCA$^{2+}$ (mg/m$^2$).*
*Results are presented for the whole year 2012 and for the temporal correlation ($R_t$), the Root Mean Squared Error (RMSE) and the bias*
*(model minus observations). The last line 'average' represents the spatial correlation $R_s$ between the mean observed and modelled values,*

| Reference | Region | $nssCA^{2+}$/dust |
|---|---|---|
| Putaud et al. (2004) | western Europe | 0.22 |
| Lequy et al. (2013) | Breuil, France | 0.03 |
| Lequy et al. (2013) | Hesse, France | 0.2 |
| This study | Africa | 0.05 (max) |
| This study | western Europe | 0.03 (max) |

**Table 8.** *Values of the ratio $nssCA^{2+}$/dust found in the literature and modeled in this study. The values correspond to the invert value of what is generally calculated.*

| AERONET stations Name | Longitude ($^o$) | Latitude ($^o$) |
|---|---|---|
| Athenes | 23.77 | 37.98 |
| Banizoumbou | 2.66 | 13.54 |
| Barcelona | 2.11 | 41.38 |
| Bastia | 9.44 | 42.69 |
| Bodele | 18.55 | 16.88 |
| Bondoukoui | -3.75 | 11.85 |
| Brussels | 4.35 | 50.78 |
| Cinzana | -5.93 | 13.27 |
| Cabauw | 4.92 | 51.97 |
| CapoVerde | -22.93 | 16.73 |
| Chilbolton | -1.43 | 51.14 |
| Dakar | -16.95 | 14.39 |
| Dahkla | -15.95 | 23.72 |
| Evora | -7.91 | 38.56 |
| ForthCrete | 25.27 | 35.31 |
| Granada | -3.60 | 37.16 |
| Ilorin | 4.34 | 8.320 |
| Izana | -16.49 | 28.31 |
| Katibougou | -7.53 | 12.92 |
| Karlsruhe | 8.42 | 49.09 |
| LaLaguna | -16.32 | 28.48 |
| Lampedusa | 12.63 | 35.51 |
| LecceUniversity | 18.11 | 40.33 |
| Leipzig | 12.43 | 51.35 |
| Lille | 3.14 | 50.61 |
| Palaiseau | 2.20 | 48.70 |
| RomeTorVergata | 12.64 | 41.84 |
| Saada | -8 | 31 |
| Tamanrasset | 5.53 | 22.79 |
| Tenerife | -16.24 | 28.47 |
| Villefranche | 7.32 | 43.68 |
| ZinderAirport | 8.98 | 13.75 |

**Table A1.** *List of the AERONET sites for the AOD measurements.*

| EMEP stations | | Longitude | Latitude |
|---|---|---|---|
| Code | Name | ($^o$) | ($^o$) |
| BY0004R | Vysokoe | 23.43 | 52.33 |
| CH0002R | Payerne | 6.94 | 46.81 |
| CH0004R | Chaumont | 6.97 | 47.04 |
| CH0005R | Rigi | 8.46 | 47.06 |
| CZ0001R | Svratouch | 16.05 | 49.73 |
| CZ0003R | Kosetice | 15.08 | 49.58 |
| DE0001R | Westerland | 8.30 | 54.92 |
| DE0004R | Deuselbach | 7.05 | 49.76 |
| DE0005R | Brotjacklriegel | 13.21 | 48.81 |
| DE0008R | Schmücke | 10.76 | 50.65 |
| DE0009R | Zingst | 12.73 | 54.43 |
| DE0044R | Melpitz | 12.93 | 51.53 |
| ES0001R | San Pablo | -4.34 | 39.54 |
| ES0005R | Noya | -8.92 | 42.72 |
| ES0006R | Mahón | 4.31 | 39.86 |
| ES0007R | Viznar | -3.53 | 37.23 |
| ES0008R | Niembro | -4.85 | 43.44 |
| ES0009R | Campisabalos | -3.14 | 41.28 |
| ES0017R | Donana | -6.33 | 37.03 |
| GB0036R | Harwell | -1.31 | 51.57 |
| GB0048R | Auchencorth | -3.24 | 55.79 |
| HR0002R | Puntijarka | 15.96 | 45.9 |
| HR0004R | Zavizan | 14.98 | 44.81 |
| HU0002R | Kpuszta | 19.58 | 46.96 |
| IT0004R | Ispra | 8.63 | 45.8 |
| LT0015R | Preila | 21.06 | 55.35 |
| ME0008R | Zabljak | 19.13 | 43.15 |
| PL0002R | Jarczew | 21.98 | 51.81 |
| PL0003R | Sniezka | 15.73 | 50.73 |
| PL0004R | Leba | 17.53 | 54.75 |
| SI0008R | Iskrba | 14.86 | 45.56 |
| SK0002R | Chopok | 19.58 | 48.93 |
| SK0004R | Stara Lesna | 20.28 | 49.15 |
| SK0006R | Starina | 22.26 | 49.05 |
| SK0007R | Topolniky | 17.86 | 47.96 |

**Table A2.** *List of the EMEP sites for the surface concentrations measurements.*