# Peer review of "Modelling the mineralogical composition and solubility of mineral dust in the Mediterranean area with CHIMERE 2017r4"

_Geoscientific Model Development, 2019_

## Referee Comment (RC1)

**Review on "Modelling the mineralogical composition and solubility of mineral dust in the Mediterranean area with CHIMERE 2017r4" by Menut et al., 2020**

The manuscript introduces a version of the CHIMERE regional chemistry-transport model, in which the mineralogical composition of dust aerosols is taken into consideration. It also presents results from an evaluation of the simulated dust mineral species and other aerosols versus observational data for the region of Europe, the Mediterranean, and North Africa, compared to a baseline simulation that takes into account only bulk dust without mineral speciation.

The explicit modeling of the mineralogical composition does not lead to an improvement of the simulated soil dust cycle or aerosol optical depth (AOD), according to the presented results. The model version with the minerals allows to explicitly simulate regional variations of $nssCa^{2+}$ and its ratio to total dust, though, which is important for biochemical cycles.

There have not been many models, regional or global ones, that consider the mineralogical composition of soil dust so far, and the presented one is a valuable addition to the few. I have a concern with respect to methodology and a few other points that should be addressed before publication, though.

1. **Section 4.4.1:** My main concern is with respect to the approach that is chosen to account for the wet-sieving bias in the database of the mineral fractions in soil by *Journet et al.* (2014), which is used as input. The authors present functions in Equation 1 to compensate for the wet-sieving bias, with the parameters chosen in a way that the values of the functions approximate the contributions of the soil clay and silt fractions to dust aerosols in Table 2a in *Scanza et al.* (2015). This may be a valid approach, principally. However, the reasoning by the authors for doing this is that they have not been able to reproduce the numbers in Table 2a by *Scanza et al.* (2015), using the equations in that paper. The authors assume anyhow that the numbers in the Table, which are approximated were correct. On what basis do they have confidence that their own formula adequately represents the contributions of the soil clay and silt fractions to the dust aerosol distribution then? That needs to be clarified.

2. **Page 4, Section 3.2, lines 107–108:** The simulations with the GOCART model that are used for the boundary limits of the CHIMERE model domain only provide total dust fluxes. How are the mineral fractions of dust treated at the boundary limits?

   Furthermore, for better reproducibility of the results from the current study, the information on what model simulations were used for the boundary limits of the model domain should be more specific than just naming on what papers they are based. Where can the simulations be accessed?

3. The authors should add a more comprehensive description of the wet deposition scheme that is used in CHIMERE to **Section 4.4.2**, since it seems to be a crucial

part for understanding the results on wet deposition, which are presented in the manuscript.

4. It will be informative, if results (figures and/or tables) for the resulting simulated volume/mass size distribution of the total dust concentration as well as the distribution of the mineral mass fractions over the dust size bins are also shown in the manuscript, even though measurements for evaluating them are not available. It still will be valuable for comparing with other models that simulate the mineralogical composition of dust aerosols. The lack of differentiation between different dust minerals with different particles densities by gravitational settling during transport may be due to the simulated size distribution.

5. **Page 16, lines 285–286, and legend of Figure 7:** The experiment for which the absolute AOD is shown in the top panel should be explicitly named there.

6. **Page 17, line 292:** The manuscript states, "Statistical scores are calculated over 32 AERONET stations. Results are presented for selected sites in Table 6".

   The phrasing is confusing. The statistical parameters are calculated and valid for the sample of all 32 stations, are they not? In what way are results shown for "selected sites" in the table?

7. **Page 18, lines 312–313:** A conclusion is stated there: "Since the precipitations are well represented in the model, it indicates that the strength of the mineral dust plumes is overestimated in the simulation."

   This conclusion is not clear to me. The model simulated wet deposition flux of $nssCa^{2+}$ has a low bias, compared to measurements. Even if the error in the simulated precipitation is not large, the error in the deposition flux can have different causes. The entire simulated dust cycle, including the wet deposition flux may be too weak, or the simulated fraction of $nssCa^{2+}$ in dust may be too small, or the wet deposition scheme that is used to calculate the deposition fluxes may not be sufficiently efficient with respect to dust tracer removal.

8. **Page 18, lines 317–320:** I do not understand what this paragraph says. Please rephrase and explain this more clearly.

9. Units of shown variables should be added to the legend of those figures and tables where they are missing.

**Typos and language issues:**

1. **Page 3, line 71:** Remove "latter".

2. **Page 4, line 107:** Replace "where" with "for which".

3. **Page 7, line 159:** Fix typo in "individual".

4. **Page 12, lines 217–218:** Replace *"M"* with *"i"* in the description of the denotations of the equations, since *"i"* stands for the individual mineral and *"M"* for the total number of the minerals.

**References**

Journet, E., Y. Balkanski, and S. P. Harrison (2014), A new data set of soil mineralogy for dust-cycle modeling, *Atmos. Chem. Phys.*, *14* (8), 3801–3816, doi:10.5194/acp-14-3801-2014.

Scanza, R. A., N. Mahowald, S. Ghan, C. S. Zender, J. F. Kok, X. Liu, Y. Zhang, and S. Albani (2015), Modeling dust as component minerals in the Community Atmosphere Model: development of framework and impact on radiative forcing, *Atmos. Chem. Phys*, *15*, 537–561, doi:10.5194/acp-15-537-2015.

---

## Referee Comment (RC2) · Anonymous Referee #2 · 11 Mar 2020

General comments

Numerical predictions of mineralogical effects on biogeochemistry and climate are highly uncertain. The authors implemented the mineralogical database to regional chemistry transport model. They confirm that this implementation does not substantially change the results of AOD, mass concentrations and deposition fluxes, following previous studies. I have some major comments to improve the paper.

Major comments

1. A fitting function to 4 data points, which were previously calculated by another fitting function, could introduce additional numerical errors. In fact, the fitting curve in Figure 3

Interactive
comment

is apparently different from that calculated by the original function. Although this would not substantially affect the results of AOD, mass concentrations and deposition fluxes, it would modulate the numerical predictions of mineralogical effects on biogeochemistry and climate. The original function should be used to avoid the error. At least, this caution should be noted in the manuscript.

2. Previous modeling studies have already implemented the mineralogical data to atmospheric chemistry transport models. The multi-model results and observational data are available over the model domain (Ito et al, 2019). Please discuss the results of the Fe solubility.

Specific comments

p.4, l.104: Please describe the method to estimate the aerosols including nitrates, ammonium and sulphates. How do you consider the effect of mineralogical composition on these aerosol formations?

Section 4: Please describe the method to estimate the mineral dust deposition flux and specify the effect of the aerosol density on the dry deposition.

p.5, l.122: To clarify a new implementation in this work, the first part should be moved before the several changes. Otherwise, please clarify the improvements from the Beegum et al. (2016), who implemented the MODIS erodibility to the CHIMERE model.

p.5, l.126: Please show the smooth function and the comparison with the measurements. How did you apply the function to different land surfaces? Please clarify the differences in the dust emissions with and without the function.

p.9, Table 3: How did you estimate the Fe solubility of 0.17% for illite? Please specify the reference, or correct the value. Please evaluate the Fe solubility with observational data over the ocean in this paper. Please clarify the differences from previous modeling studies in the estimate of the Fe solubility.

p.10, l.177: How did you calculate the deposition fluxes for the mineral species?
p.10, l.193: Presumably, you used different size distribution of emitted aerosols. How did you calculate it? Please specify your calculation using their equation, which should provide the results presented in their Table 2a in your case.

p.16, l.284 and p.21, l.344: Please show the results of radiation effect, or rephrase the sentences.

p.21, l.333 and Table 8: Please show the range of latitude and longitude for each region and compare the results over the same region.

p.21, l.349: Please clarify the strong dependency of settling velocity on the density in the method, or rephrase the sentence.

Technical comments

Figures 1 and 2 as well as Tables 1 and 2 may be moved to supplementary materials to avoid the redundancy.

p.1, l.4: Please correct in.

p.2, l.47: Please correct out.

p.3, l.49: Please correct et.

p.5, l.113: Please remove the.

p.10, l.184: Please delete ,.

p.11, Figure 3: Please correct f_clay [Scanza] and weight functions.

p.12, l.212: Please correct emission.

p.12, l.216: Please add the number to the equation.

P.17, Figure 7: Please correct the caption.

References

Beegum, S., I. Gherboudj, N. Chaouch, F. Couvidat, L. Menut, and H. Ghedira: Simulating Aerosols over Arabian Peninsula with CHIMERE: Sensitivity to soil, surface parameters and anthropogenic emission inventories, Atmospheric Environment, 128, 185–197, https://doi.org/10.1016/j.atmosenv.2016.01.010, 2016.

Ito, A., Myriokefalitakis, S., Kanakidou, M., Mahowald, N. M., Scanza, R. A., Hamilton, D. S., Baker, A. R., Jickells, T., Sarin, M., Bikkina, S., Gao, Y., Shelley, R. U., Buck, C. S., Landing, W. M., Bowie, A. R., Perron, M. M. G., Guieu, C., Meskhidze, N., Johnson, M. S., Feng, Y., Kok, J. F., Nenes, A., and Duce, R. A.: Pyrogenic iron: The missing link to high iron solubility in aerosols, Sci. Adv., 5, eaau7671, https://doi.org/10.1126/sciadv.aau7671, 2019.

———————————————————

---

## Author Comment (AC1) · 20 Mar 2020

**Modelling the mineralogical composition and solubility of mineral dust in the Mediterranean area with CHIMERE 2017r4**

Menut, L., Siour, G., Bessagnet, B., Couvidat, F., Journet, E., Balkanski, Y., and Desboeufs, K. https://www.geosci-model-dev-discuss.net/gmd-2019-337/

Dear Editor and reviewers,

We acknowledge the reviewers for the time spent to evaluate our work and for their minor revisions. We also acknowledge the Editor and we made all proposed changes in the revised manuscript. Please note that answers are in blue and after each reviewer's remark. When a large paragraph is added in the manscript, it is here described in a grey box.

All reviewers remarks were taken into account and are detailed in this letter. Summarizing our answers:

- 1. Text, references and Figures (captions and labels) were checked and corrected as requested.
- 2. The two reviewers have questions about the function proposed to estimate the relative ratio of silt and clay as a function of the mean mass median diameter of the aerosol. We present here the problem 15 we had: the goal of this function is to provide a simple and smooth transition between silt and clay fraction. The function proposed by Scanza et al. (2015) is very complex and when we computed it, we did not find the values presented in their article. Thus, we prefer to calculate this transition using another function, more simple and providing the same values.
- 3. The two reviewers ask for more details about the dry and wet deposition schemes used in the model. 20 We add a section describing in detail these calculations.

Best regards, Laurent Menut March 20, 2020 5

**1 Reviewer #1**

**Received and published: 20 February 2020**

- 5 The manuscript introduces a version of the CHIMERE regional chemistry-transport model, in which the mineralogical composition of dust aerosols is taken into consideration. It also presents results from an evaluation of the simulated dust mineral species and other aerosols versus observational data for the region of Europe, the Mediterranean, and North Africa, compared to a baseline simulation that takes into account only bulk dust without mineral speciation.
- 10 The explicit modeling of the mineralogical composition does not lead to an improvement of the simulated soil dust cycle or aerosol optical depth (AOD), according to the presented results. The model version with the minerals allows to explicitly simulate regional variations of nssCa2+ and its ratio to total dust, though, which is important for biochemical cycles.

There have not been many models, regional or global ones, that consider the mineralogical composition of 15 soil dust so far, and the presented one is a valuable addition to the few. I have a concern with respect to methodology and a few other points that should be addressed before publication, though.

- 1. Section 4.4.1: My main concern is with respect to the approach that is chosen to account for the wetsieving bias in the database of the mineral fractions in soil by Journet et al. (2014), which is used as input. The authors present functions in Equation 1 to compensate for the wet-sieving bias, with
- the parameters chosen in a way that the values of the functions approximate the contributions of the soil clay and silt fractions to dust aerosols in Table 2a in Scanza et al. (2015). This may be a valid approach, principally. However, the reasoning by the authors for doing this is that they have not been able to reproduce the numbers in Table 2a by Scanza et al. (2015), using the equations in that paper. The authors assume anyhow that the numbers in the Table, which are approximated were correct. On what basis do they have confidence that their own formula adequately represents the contributions of
- the soil clay and silt fractions to the dust aerosol distribution then? That needs to be clarified.

**Answer:**

30

35

45

We understand this reviewer's remark. Probably, it was not well written in our manuscript. The problem was that we used the equation and were not able to calculate the same values as in table 2. The equation in Scanza et al. (2015) provides unrealistic values. This is why we preferred to write another equation able to provide values close to the ones in Table 2. This is illustrated in the answer to the Reviewer #2. About our confidence in the result of this simplified equation, we show in Figure 3 a comparison between the values of Scanza et al. (2015) and our equation: when Scanza et al. (2015) only provides mean values per size intervals, we propose a continuous fit, then not dependent on the model size distribution. And for all aerosol sizes, our fit gives the same values than the constant values of Scanza et al. (2015).

2. Page 4, Section 3.2, lines 107-108: The simulations with the GOCART model that are used for the boundary limits of the CHIMERE model domain only provide total dust fluxes. How are the mineral fractions of dust treated at the boundary limits?

40 Answer:

The reviewer is right, this problem was one of our major problem when developing the mineralogy in CHIMERE: how to redistribute the mineral dust at the boundaries without knowing their origin then their mineralogy? In fact, there is no rigourous answer and we then decided to create an 'other' category where we put initial and boundary concentrations. This is not very clean but it is without error. After tests, it was shown that this contribution is negligible compared to the emissions calculated in the modelled domain. This was added in the manuscript. The section 4.4 is in the manuscript about this point:

In addition, one has to note that the boundary conditions for mineral dust are entirely assigned to the "other" species, DuOT. Indeed, having no information from the global model used for the boundary conditions, it was not possible to assign these concentrations to specific minerals. To

minimize the impact of this approximation on the boundary conditions, the simulations used in this study are done over a large domain.

Furthermore, for better reproducibility of the results from the current study, the information on what model simulations were used for the boundary limits of the model domain should be more specific than just naming on what papers they are based. Where can the simulations be accessed?

**Answer:**

The study was done with these GOCART boundary conditions and, apart the problem well pointed out by the reviewer, we are aware it is an old climatology, with a low resolution and too few informations. Our goal, for the next CHIMERE version, is too change these boundary conditions with a more recent dataset such as the CAMS global data. But, to be more accurate in this article, we added a new subsection called "Boundary conditions" containing:

"Boundary conditions for mineral dust are calculated using a climatology calculated with the GOCART model, (Ginoux et al., 2001). This climatology was provided by Mian Chin and Paul Ginoux for the CHIMERE validation and distribution to users. The data are freely available on the CHIMERE download web site. The data represent a monthly global climatology simulation of mineral dust with an horizontal resolution of  $2.5^{\circ} \times 2^{\circ}$ , Ginoux et al. (2001). The monthly mineral dust concentration fields are an averaged of years 1987, 1988, 1989, 1990 and 1997 and proposed in 7 size bins, later reprojected in the CHIMERE aerosol bins. The use of this climatology in the case of this study has a major weakness, knowing that the data are for the usual mean mineral dust species. The question was thus to choose how to redistribute this mean species into all mineralogical species. To avoid errors, it was decided to add this contribution into the 'other' species called DuOT. After some test cases, it was shown that the contribution of the boundary conditions was very low compared to the emissions calculated in the modelled domain. The impact of this hypothesis on the results was found to be negligible."

3. The authors should add a more comprehensive description of the wet deposition scheme that is used in CHIMERE to Section 4.4.2, since it seems to be a crucial part for understanding the results on wet deposition, which are presented in the manuscript.

**Answer:**

Additional informations about the dry and wet deposition fluxes calculation were added in the section 4 (see answer to Reviewer #2 where the whole text is extensively provided). This section was renamed because contains more informations than only about emissions. In place of "The mineral dust emissions", it is now "Model changes for mineral dust mineralogy". And it contains all changes made in the 20 CHIMERE v2017r4 to include the mineralogy calculations. A new subsection was also added called "Boundary conditions" to better answer the previous remark of reviewer #1.

4. It will be informative, if results (figures and/or tables) for the resulting simulated volume/mass size distribution of the total dust concentration as well as the distribution of the mineral mass fractions over the dust size bins are also shown in the manuscript, even though measurements for evaluating them are not available. It still will be valuable for comparing with other models that simulate the mineralogical composition of dust aerosols. The lack of differentiation between different dust minerals with different particles densities by gravitational settling during transport may be due to the simulated size distribution.

**Answer:**

Yes, that's right. The goal of such study is to have more detail about the mineralogical and chemical composition of the mineral dust. We already provide this information with the Figure 4: monthly mean surface concentrations are presented for several sites, representative of different regions and with the mineralogical composition. We think it makes sense with this format. The conclusion of this figure

5

15

was that the variability remains very close to the emission and the emission of the different minerals was not so variable in space. Figure 4 is thus really representative of the result we could obtain and we concluded that to present the same kind of composition but with another choice of plot would be not very different.

5. Page 16, lines 285-286, and legend of Figure 7: The experiment for which the absolute AOD is shown in the top panel should be explicitly named there.

Answer:

Yes, of course, this information was missing. The absolute value of AOD is shown for the 'DUST" simulation. It was added in the text and in the Figure 7 caption. The new text is:

The top panel displays the mean averaged value of AOD for April 2012 (when the largest surface concentrations were modelled) and for the simulation 'DUST'.

**The corrected caption is:**

Monthly averaged Aerosol Optical Depth for April 2012 and over the whole modeled domain. (top) AOD absolute values are presented for the simulation DUST. (bottom) The map of difference represents the calculation of AOD(DUST)-AOD(MNRLO).

6. Page 17, line 292: The manuscript states, 'Statistical scores are calculated over 32 AERONET stations. Results are presented for selected sites in Table 6'. The phrasing is confusing. The statistical parameters 15 are calculated and valid for the sample of all 32 stations, are they not? In what way are results shown for 'selected sites' in the table?

**Answer:**

The text was wrong and was corrected. The statistical scores are calculated over 32 AERONET stations and there is no 'selected sites' in Table 6.

7. Page 18, lines 312-313: A conclusion is stated there: 'Since the precipitations are well represented in the model, it indicates that the strength of the mineral dust plumes is overestimated in the simulation.

This conclusion is not clear to me. The model simulated wet deposition flux of  $nssCa^{2+}$  has a low bias, compared to measurements. Even if the error in the simulated precipitation is not large, the error in the deposition flux can have different causes. The entire simulated dust cycle, including the wet deposition flux may be too weak, or the simulated fraction of  $nssCa^{2+}$  in dust may be too small, or the wet deposition scheme that is used to calculate the deposition fluxes may not be sufficiently efficient with respect to dust tracer removal.

**Answer:**

We are OK with this remark: our conclusion was too fast and too definitive. The reviewer is right: a 30 lot of different processes may be at the origin of this underestimation. We change the sentence to open more to several possible causes. The sentence is then replaced by:

> The capability of a model to simulate the dust cycle contains many processes and, then, many possible errors. Here, the precipitation is correctly represented by the model. It means that the underestimation of modelled wet deposition fluxes compared to the measurements is probably due to other processes than a misrepresentation of the precipitation. This could be the altitude of the precipitating clouds, the trajectory of dust plumes (missing a station or not), the efficiency of the parameterized scavenging, possible errors on dust size distribution, a too small simulated fraction of nssCa2+, among other possibilities.

5

10

20

8. Page 18, lines 317-320: I do not understand what this paragraph says. Please rephrase and explain this more clearly.

**Answer:**

Yes, it was an attempt to explain that the biases should be quite spatialized but are not spatialized at all in the end. Indeed, it is not clear. But as it is just a general idea, but nothing very accurate, we prefer remove this part from the manuscript. The paragraph is replaced by:

Since dust plumes are very spatially extensive, there is usually a bias between model and measurements for groups of stations located beneath these plumes. This is not the case here, since there are highly variable biases for nearby stations. The origin of the bias is therefore not due to a 'large scale' error: it is therefore probably not a transport problem. But it may be a precipitation problem, which is often a phenomenon of greater spatial variability on a small scale.

9. Units of shown variables should be added to the legend of those figures and tables where they are missing.

**Answer: That has been corrected for all Figures and Tables where it was the case.**

**Typos and language issues:**

- 1. Page 3, line 71: Remove 'latter'.
- 2. Page 4, line 107: Replace 'where' with 'for which'.
- 3. Page 7, line 159: Fix typo in 'individual'.
- 4. Page 12, lines 217-218: Replace 'M' with 'i' in the description of the denotations of the equations, since 'i' stands for the individual mineral and 'M' for the total number of the minerals.

**Answer:**

The description of equation 5 was corrected and is now:

$$\begin{cases} DF_{Nso}^{b} = \sum_{i=1}^{M} \left( DF_{i}^{b} \times \mathcal{H}_{Ni} \times \mathcal{H}_{solubNi} \right) \\ DF_{Nin}^{b} = \sum_{i=1}^{M} \left( DF_{i}^{b} \times \mathcal{H}_{Ni} \times (1 - \mathcal{H}_{solubNi}) \right) \end{cases}$$
(1)

with  $DF_N^b$  the deposition flux of the chemical element N (i.e. Fe, Ca etc.) for its bin b. This flux is splitted between the soluble (so) and insoluble (in) parts.  $\mathscr{H}_{Ni}$  is the percentage of chemical element N in each mineral M,  $\mathscr{H}_{solubNi}$  the percentage of soluble fraction of chemical element N in each mineral M.

Please note that the number was missing for this equation and was added for the revised version. Note also that we replaced nb by b for the bin number to avoid confusion with "number".

**Answer: That has been corrected for all remarks.**

10

15

**References**

- Ginoux, P., Chin, M., Tegen, I., Prospero, J. M., Holben, B., Dubovik, O., and Lin, S. J.: Sources and distributions of dust aerosols simulated with the GOCART model, Journal of Geophysical Research, 106, 20255–20273, 2001.
- 5 Journet, E., Balkanski, Y., and Harrison, S. P.: A new data set of soil mineralogy for dust-cycle modeling, Atmospheric Chemistry and Physics, 14, 3801–3816, https://doi.org/10.5194/acp-14-3801-2014, 2014.
- Scanza, R. A., Mahowald, N., Ghan, S., Zender, C. S., Kok, J. F., Liu, X., Zhang, Y., and Albani, S.: Modeling dust as component minerals in the Community Atmosphere Model: development of framework and impact on radiative forcing, Atmospheric Chemistry and Physics, 15, 537–561, https://doi.org/10.5194/acp-15-537-2015, 2015.

---

## Author Comment (AC2)

**Modelling the mineralogical composition and solubility of mineral dust in the Mediterranean area with CHIMERE 2017r4**

Menut, L., Siour, G., Bessagnet, B., Couvidat, F., Journet, E., Balkanski, Y., and Desboeufs, K.
https://www.geosci-model-dev-discuss.net/gmd-2019-337/      5

Dear Editor and reviewers,

We acknowledge the reviewers for the time spent to evaluate our work and for their minor revisions. We also acknowledge the Editor and we made all proposed changes in the revised manuscript. Please note that answers are in blue and after each reviewer's remark. When a large paragraph is added in the manscript, it is here described in a grey box.      10
All reviewers remarks were taken into account and are detailed in this letter.
Summarizing our answers:

1. Text, references and Figures (captions and labels) were checked and corrected as requested.

2. The two reviewers have questions about the function proposed to estimate the relative ratio of silt and clay as a function of the mean mass median diameter of the aerosol. We present here the problem   15 we had: the goal of this function is to provide a simple and smooth transition between silt and clay fraction. The function proposed by Scanza et al. (2015) is very complex and when we computed it, we did not find the values presented in their article. Thus, we prefer to calculate this transition using another function, more simple and providing the same values.

3. The two reviewers ask for more details about the dry and wet deposition schemes used in the model.   20 We add a section describing in detail these calculations.

Best regards,
Laurent Menut
March 20, 2020

**1   Reviewer #2**

**General comments**

Numerical predictions of mineralogical effects on biogeochemistry and climate are highly uncertain. The authors implemented the mineralogical database to regional chemistry transport model. They confirm that this implementation does not substantially change the results of AOD, mass concentrations and deposition fluxes, following previous studies. I have some major comments to improve the paper.

**Major comments**

1. A fitting function to 4 data points, which were previously calculated by another fitting function, could introduce additional numerical errors. In fact, the fitting curve in Figure 3 is apparently different from that calculated by the original function. Although this would not substantially affect the results of AOD, mass concentrations and deposition fluxes, it would modulate the numerical predictions of mineralogical effects on biogeochemistry and climate. The original function should be used to avoid the error. At least, this caution should be noted in the manuscript.

*Answer:*
These remarks, and the associated doubt, about this function we proposed is a common point between the two reviewers. Our goal was primarily to use the Scanza et al. (2015) proposed function in our model. But results were not correct. We double checked the code (L. Menut and G. Siour) and found that we coded exactly what is in the paper. As, in addition, the formulation was very complicated in regard of the searched goal (i.e only provide a smooth transition from silt and clay, with values from 0 to 1), we preferred to implement a more simple and robust equation. We consider this is not adding potential errors as the reviewer write: the main idea of the authors in Scanza et al. (2015) is to propose a smooth transition but the absolute values they proposed are not exact and are proposed with constant values in large intervals.

[Figure]

**Figure 1.** *Comparison between the function proposed by Scanza et al. (2015) and the one proposed in this study. The formulation by Scanza et al. (2015) is complex and appears to be numerically unstable with false values: negative values appear when we want only a smooth factor from 0 to 1. In addition, some unrealistic peaks are present, not conserving the total of 1 with the two parts of the function.*

Figure 1 displays a comparison between the Scanza et al. (2015) equation and the one we proposed in this article. For mean mass median diamters below $1\mu m$, it is clear that the Scanza et al. (2015) has unrealistic peak and negative values, when we expect to have a transition factor between 0 and 1. It is

delicate to write in an article that the function we want to use is not correct and it is why we did not in the first version of the manuscript. But since the two reviewers have doubts about this, we modified this sentence in the manuscript:

> The relative part of clay and silt for each mineral depends on the mean mass median diameter of the emitted aerosol. We attempted to follow the formulation proposed in Scanza et al. (2015), with an equation and corresponding results in a Table. Unfortunately, the coding of the proposed formulation provides erroneous values, largely different from the results presented in their Table. Their formulation appears to be numerically not correct, in any case far from the simple goal which is to have a factor giving a smooth transition between 0 and 1. We thus define a new and simplified formulation as:...

2. Previous modeling studies have already implemented the mineralogical data to atmospheric chemistry transport models. The multi-model results and observational data are available over the model domain (Ito et al, 2019). Please discuss the results of the Fe solubility.

   *Answer:*
   Thanks for the reference of Ito et al. (2019) (note this paper was not published when our work was done and the draft written, then submitted). Then a discussion based on this multi-model work was added in the manuscript. About the Fe solubility, we are not in the same configuration than the models used in this study. For the moment, and in our model, there is no Fe in our anthropogenic emissions, thus it is not realistic to discuss the Fe solubility neither than compare the concentrations to measurements. But we added a sentence about this interesting point in the conclusion. This sentence was added at the end of the manuscript:

   > A step forward could be to add the Fe anthropogenic emissions in the model and then to have realistic Fe concentrations and deposited fluxes to make comparisons to measurements as done for example in Ito et al. (2019).

**Specific comments**

- p.4, l.104: Please describe the method to estimate the aerosols including nitrates, ammonium and sulphates. How do you consider the effect of mineralogical composition on these aerosol formations?

  *Answer:*
  In the first version of this manuscript, there was no description of active chemistry because the paper is dedicated to mineral dust. But CHIMERE is able to model inorganic species such as nitrates, ammonium and sulphates. It would be very long and not very useful to describe it in this 'dust' paper. Thus, we added references about the processes and chemical species. Note also that, for the moment, there is no link in the model between mineral dust and chemically active species. In this model version, mineral dust are considered as chemically inert. But these interactions could be the goal of future developments.
  The following sentence was added in the manuscript:

  > A complete chemistry is included in the model, a general description of gaseous and aerosol schemes is provided in Mailler et al. (2017) for this model version, including a detailed description of the aerosol scheme in Couvidat et al. (2018). The chemical evolution of gaseous species is calculated using the MELCHIOR2 scheme. The aerosol size distribution is represented using ten bins, from 40 nm to 40 $\mu$m, in mean mass median diameter as described in Menut et al. (2016) and updated in Mailler et al. (2017).

- Section 4: Please describe the method to estimate the mineral dust deposition flux and specify the effect of the aerosol density on the dry deposition.

*Answer:*
Yes, this request was also written by Reviewer #1. A subsection was added in the manuscript, describing the dry and wet deposition schemes used for aerosols. This new section is as follows:

[revised manuscript text omitted]

with $\lambda$ the mean free path of air, in meters, estimated as:

$$\lambda = \frac{2\mu_{\text{air}}}{p\sqrt{\frac{8M_{\text{air}}}{\pi RT}}} \tag{12}$$

where $M_{\text{air}}$ is the molecular mass of dry air (here 28.8 g mol$^{-1}$), $T$ the temperature (K), $p$ the pressure (Pa), $\mu$ the air dynamic viscosity and $R$ the universal gas constant.

*Answer:*

This also answers the question about the use of the density for the dry deposition. The wet deposition is 5 calculated as follows:

The aerosols wet deposition calculation is separated between rain and snow. There is also a distinction between the wet deposition in-cloud and below cloud.
For below-cloud scavenging, aerosols are scavenged by raining drops. Following Willis and Tattelman (1989), a polydisperse distribution of raining drops is applied:

$$N(R) = 1.06 \times 10^{14} \times P^{-0.0295d0}(2R)^{2.16d0}\exp\left(-5679P^{-0.153d0}2R\right) \tag{13}$$

with $P$ the precipitation rate in mm/h and $R$ the radius of the droplet (in m).
The flux of deposition is calculated with:

$$F_{bc}^i = c^i \times \sum_R \pi R^2 u_g(R) E(R, r_i) N(R) \tag{14}$$

with $i$ the aerosol species, $r_l$ the radius of the particle (in m), $u_g$ the terminal drop velocity (in m/s), $E(R, r_l)$ the collision efficiency of a particle with a raindrop, $N(R)$ (in m$^{-4}$) the raindrop size distribution.
For below-cloud scavenging of particles by snow, the particles are scavenged by appling the parameterization of Wang et al. (2014). A scavenging coefficient $\lambda_{snow}$ is computed with:

$$log\,(\lambda_{snow}) = log\,A + B \tag{15}$$

with $A$ and $B$ fit function depending on the aerosol mean mass median diameter $D_p$. The flux of deposition is calculated with:

$$F_{in}^i = -\lambda_{snow} \times c^i \tag{16}$$

For in-cloud scavenging is here considered only when a precipitation occurs. The rate of deposition is computed by calculating the rate of impaction between hydrometeors and cloud droplets (assumed to have a diameter of 10 $\mu$m). The rate of scavenging is computed with equations 14 and 16 for $D_p$=10$\mu$m.

- p.5, l.122: To clarify a new implementation in this work, the first part should be moved before the several changes. Otherwise, please clarify the improvements from the Beegum et al. (2016), who implemented the MODIS erodibility to the CHIMERE model.

  *Answer:*
  Following this remark, the sections were redesigned. One part for the distributed CHIMERE version and another separated part for the new implementation related to dust mineralogy. Many details are already presented in the publication of Beegum et al. (2016) about erodibility and we added a few details.

- p.5, l.126: Please show the smooth function and the comparison with the measurements. How did you apply the function to different land surfaces? Please clarify the differences in the dust emissions with and without the function.

  *Answer:*
  The smooth function was already showed and explained in Mailler et al. (2017). As it is not authorized (and useful) to publish two times the same thing, we added a reference to the original publication. About the different landuses, there is no use of this information. This was added in the text.

- p.9, Table 3: How did you estimate the Fe solubility of 0.17% for illite? Please specify the reference, or correct the value. Please evaluate the Fe solubility with observational data over the ocean in this paper. Please clarify the differences from previous modeling studies in the estimate of the Fe solubility.

  *Answer:*
  The values are extracted from the Paris et al. (2011) work. Since we used exactly the values already discussed in this paper, we added the reference directly in the section 4.3 and Table 3. Our goal is not to evaluate the chosen constants in this paper since it was already done in other papers with exactly the same values, such as Balkanski et al. (2007). We remind that this study is dedicated to implement in the regional model CHIMERE exactly the same type of calculation than in the LMDz model: the goal is to have this functionality in a regional model in addition to the global one.

- p.10, l.177: How did you calculate the deposition fluxes for the mineral species?

*Answer:*

This question was already adressed and then already answered. Please see the answer to your first comment.

- p.10, l.193: Presumably, you used different size distribution of emitted aerosols. How did you calculate it? Please specify your calculation using their equation, which should provide the results presented in their Table 2a in your case.

*Answer:*

There is several questions in this comment. About the question on the calculation of the Scanza et al. (2015) formulation, at the beginning of this answer, we already presented the result of our calculation, checked many times and the Reviewer can see the problem. About the calculation of the aerosol size distribution used for the transport and then applied in equation (1), it is fully described in Mailler et al. (2017). A sentence was added about this point is the section about the model presentation.

> The aerosol size distribution is represented using ten bins, from 40 nm to 40 $\mu$m, in mean mass median diameter as described in Menut et al. (2016) and updated in Mailler et al. (2017).

- p.16, l.284 and p.21, l.344: Please show the results of radiation effect, or rephrase the sentences.

*Answer:*

The term 'radiation' was used to say that a change in aerosol optical properties may induce a change in radiation and it is quantifiable via the AOD. The sentence was changed to eliminate 'radiation'. In the conclusion, it is correct since a change of the refractive index has an impact on the aerosol radiative effects.

- p.21, l.333 and Table 8: Please show the range of latitude and longitude for each region and compare the results over the same region.

*Answer:*

The range in latitude is expressed directly with the "Region" column. For the results of Lequy et al. (2013), there is no range since it corresponds to measurements at a specific location. The idea behind this table was to propose a discussion about the very large variability of this ratio and to show that very few studies were dedicated to its quantification. In our study, we propose a map and it is a novelty. Restricting the comparison by latitude bands is not at all the purpose of the table presented.

- p.21, l.349: Please clarify the strong dependency of settling velocity on the density in the method, or rephrase the sentence.

*Answer:*

This question was already adressed by this Reviewer and we answered with the complete description (as requested) of the dry and wet deposition calculation.

**Technical comments**

- Figures 1 and 2 as well as Tables 1 and 2 may be moved to supplementary materials to avoid the redundancy.

*Answer:*

These Tables are really useful since they concatenate all interesting informations about the choices made to represent the mineralogy in a model. We think they are their place in the article. The Figures are here to show the differences between the values and it is a complementary information that the quantified values in the Tables.

- p.1, l.4: Please correct in.

*Answer:*
Corrected.

- p.2, l.47: Please correct out.

  *Answer:*
  Corrected.

- p.3, l.49: Please correct et.

  *Answer:*
  We don't see the problem with line. But the text was corrected in a general manner.

- p.5, l.113: Please remove the.

  *Answer:*
  Corrected. The complete paragraph was rewritten.

- p.10, l.184: Please delete ,.
- p.11, Figure 3: Please correct f_clay [Scanza] and weight functions.

  *Answer:*
  Yes, right. The Figure was reprocessed to correct this error. The new Figure is:

[Figure]

- p.12, l.212: Please correct emission.

  *Answer:*
  Corrected.

- p.12, l.216: Please add the number to the equation.

  *Answer:*
  Corrected.

- P.17, Figure 7: Please correct the caption.

  *Answer:*
  Corrected. The caption is now:

[revised manuscript text omitted]